# Carbon Nanodot–Microbe–Plant Nexus in Agroecosystem and Antimicrobial Applications

**DOI:** 10.3390/nano14151249

**Published:** 2024-07-25

**Authors:** József Prokisch, Duyen H. H. Nguyen, Arjun Muthu, Aya Ferroudj, Abhishek Singh, Shreni Agrawal, Vishnu D. Rajput, Karen Ghazaryan, Hassan El-Ramady, Mahendra Rai

**Affiliations:** 1Faculty of Agricultural and Food Sciences and Environmental Management, Institute of Animal Science, Biotechnology and Nature Conservation, University of Debrecen, 138 Böszörményi Street, 4032 Debrecen, Hungary; nguyen.huu.huong.duyen@agr.unideb.hu (D.H.H.N.); arjun.muthu@agr.unideb.hu (A.M.); ferroudj.aya@agr.unideb.hu (A.F.); mahendrarai@sgbau.ac.in (M.R.); 2Tay Nguyen Institute for Scientific Research, Vietnam Academy of Science and Technology (VAST), Dalat 66000, Vietnam; 3Doctoral School of Nutrition and Food Science, University of Debrecen, 138 Böszörményi Street, 4032 Debrecen, Hungary; 4Doctoral School of Animal Husbandry, University of Debrecen, 138 Böszörményi Street, 4032 Debrecen, Hungary; 5Faculty of Biology, Yerevan State University, Yerevan 0025, Armenia; sinxabishik@ysu.am (A.S.); kghazaryan@ysu.am (K.G.); 6Department of Biotechnology, Parul Institute of Applied Science, Parul University, Vadodara 391760, Gujarat, India; shreniagrawal01@gmail.com; 7Academy of Biology and Biotechnology, Southern Federal University, Rostov on Don 344006, Russia; rvishnu@sfedu.ru; 8Soil and Water Department, Faculty of Agriculture, Kafrelsheikh University, Kafr El-Sheikh 33516, Egypt; 9Department of Biotechnology, Sant Gadge Baba Amravati University, Amravati 444602, Maharashtra, India

**Keywords:** soil–plant–microbe nexus, microbiomes, rhizosphere, microbe–microbe nexus, nanoparticles, nanotoxicity

## Abstract

The intensive applications of nanomaterials in the agroecosystem led to the creation of several environmental problems. More efforts are needed to discover new insights in the nanomaterial–microbe–plant nexus. This relationship has several dimensions, which may include the transport of nanomaterials to different plant organs, the nanotoxicity to soil microbes and plants, and different possible regulations. This review focuses on the challenges and prospects of the nanomaterial–microbe–plant nexus under agroecosystem conditions. The previous nano-forms were selected in this study because of the rare, published articles on such nanomaterials. Under the study’s nexus, more insights on the carbon nanodot–microbe–plant nexus were discussed along with the role of the new frontier in nano-tellurium–microbe nexus. Transport of nanomaterials to different plant organs under possible applications, and translocation of these nanoparticles besides their expected nanotoxicity to soil microbes will be also reported in the current study. Nanotoxicity to soil microbes and plants was investigated by taking account of morpho-physiological, molecular, and biochemical concerns. This study highlights the regulations of nanotoxicity with a focus on risk and challenges at the ecological level and their risks to human health, along with the scientific and organizational levels. This study opens many windows in such studies nexus which are needed in the near future.

## 1. Introduction

The agroecosystem is the main ecosystem in which plants, animals, and other organisms can interact together to produce the needed food, feed, fiber, and fuel for human life. This system is very important for our lives and controls the development and sustainability of the agriculture sector. Climate change and high food prices are driving food and nutrition insecurity, pushing millions into extreme poverty, and reversing hard-won development gains. Around a quarter of a billion people now face acute food insecurity. Millions of people are either not eating enough or eating the wrong types of food, resulting in a double burden of malnutrition that can lead to illnesses and health crises. Agricultural development is one of the most powerful tools to end extreme poverty, boost shared prosperity, and feed a projected 10 billion people by 2050. Growth in the agriculture sector is two to four times more effective in raising incomes among the poorest compared to other sectors. Agriculture is also crucial to economic growth: accounting for 4% of global gross domestic product (GDP) and in some least developing countries, it can account for more than 25% of GDP (https://www.worldbank.org/ [1]). This potential is very clear from different reviews published recently with a focus on many topics such as agroecosystem services [2], microbiomes in agroecosystems [3], nano-plastics in agroecosystems [4,5], pest suppressive agroecosystems [6], vegetation in the Canadian prairie [7], microbial N-management [8], engineering of the rhizosphere [9], socio-economic agroecology [10], nano-pollution of agroecosystems [11,12], and exploration the antimicrobial resistance in agroecosystem [13]. Thus, engineering, designing, and managing the agroecosystem is a crucial issue through the study of the interaction among the compartments of soil, microbes, and plants [6].

The relationship between agriculture and nanotechnology is not new; rather, it has existed for a very long time since agricultural soil naturally contains nanoparticles and other nanomaterials. The penetration of nanotechnology applications into all agricultural practices has further complicated this situation [14,15] along with many global issues such as agri-food industries [16], agri-food waste valorization [17], global food insecurity under climate change [18], proliferating agriculture sector [19], and sustainability [20,21]. The nanomaterial–microbe–plant nexus has received more concern in recent decades due to its potential in the agriculture sector [22]. Under agroecosystems, the fate and transport of nanoparticles (NPs) in soil and plant systems could be considered crucial and mediated by different microbial communities [23,24,25]. The microbial role in NP immobilization and degradation in agroecosystems also is an important issue under such a nexus [26]. Thus, this nexus opens several windows regarding the crucial interactions among soil, plant, microbes, and NPs under agroecosystems.

Therefore, this review focuses on the different suggested interactions among the NP–microbe–plant nexus in agroecosystems. This study also will discuss the possible transport, nanotoxicity, and regulations of NPs along with the perspectives, and challenging gaps.

## 2. Microbes and Plants: Amazing World

Under the agroecosystem, the plant–microbe interaction can be noticed as a dynamic, complex, and continuous process besides its being an old colonization of plants on the Earth [27]. The association between plants and microbes started from billions with the stromatolite formation around 3.5 billion years ago and can be found in many forms such as microbe–microbe or/and plant–microbe interactions [28,29,30], microbes for phytoremediation of polluted soil or water or air [31,32,33], endosymbiosis or plant growth-promoting micro-organisms [34], microbes on rhizoplane/rhizosphere [35], and microbes live between plant cells or endophytes [36]. These relationships were discussed from different points of view including the following issues:Producing small peptides by microbes and/or plants and their role in plant–microbe interactions in the rhizosphere along with the change in the rhizosphere microbiomes. This nexus may be also useful in holobiont engineering, and the potential of exploring transgenic microbes to synthesize small peptides on a large scale [37].The role of plant–microbe interactions in regulating sterols including phytosterol biosynthesis, recognition, communication, transduction, and/or exchanges between partners through the expression of genes [38].The plant–microbe interactions and their impacts on polluted soil with microplastics through the microbial degradation microbial-mediated MPs bioremediation [39].Genome of the plant–microbe interactions under different ecological, physiological, and evolutionary implications of both plants and microbes and their impacts on stress tolerance, growth, and nutrient acquisition [40].The plant–microbe interactions and their role in controlling crop productivity through impacting the microbial communities, and activities, influencing endogenous and external growth factors, and potential targeted applications in agricultural production [41].The role of plant–microbe interactions in transferring immune signals between plant cells and plant pathogens including different bioactive molecules (as extracellular vesicles) like metabolites, proteins, lipids, and small RNAs and facilitating the exchange of such active substances between various species [40].Soil microbes and their play in nutrient cycling, soil health, and ecological restoration for food security and nutrient quality, which are controlled by both plants and microbes [42].The specific role of plant–metabolite–microbe or pathogen interactions and their complexity for the identification of specialized metabolite pathways using ecological, mechanistic, and evolutionary models [43].The role of agricultural practices (mainly plant grafting) in increasing or suppressing crop productivity, alleviating abiotic stress, controlling pathogens, and modulating the root microbiome [44].The interaction among nanoparticles, plants, and microbes and their roles in crop production and food security through the use of nano-devices/-products for agro-applications [45].

## 3. Relation between Microbes and Plants under Different Soil Conditions

What is the possible nexus between plants and microbes under different soil conditions? What are the main factors controlling this relationship? Soil conditions and properties are essential in impacting the plant–microbe nexus. These conditions may include polluted soils which are needed for microbe–plant assisted bioremediation [46], the intercropping system [47], applying soil organic amendments for a long time [48], soil weed rhizosphere phenolics [49], the relation among plant, soil, microbe, and anthropogenic activities for soil health [50], myco-plant remediation of polluted soils [51], and different types of soil cover [52]. Rhizosphere soil properties are the main controller and guarantee of the plant—microbe nexus [53], which may be related to the characterization of cropping systems, and other agro-practices (Table 1).

## 4. Nanomaterial–Microbe–Plant Nexus

### 4.1. Carbon Nanodots as Inhibitors of Phytopathogens

Microbes play an important role in agroecosystems, as they are involved in nutrient cycling, decomposition, and disease suppression. Carbon nanodots (CNDs) have the potential to be used in agroecosystems to control microbial populations. For example, CNDs could be used to kill or inhibit the growth of harmful bacteria, such as those that cause plant diseases. Additionally, CNDs could be used to promote the growth of beneficial bacteria, such as those that fix nitrogen or solubilize phosphorus. Research has shown that CNDs, a type of carbon nanomaterial, possess significant antimicrobial properties for various microbes [60]. Table 2 shows that CNDs derived from various sources have antimicrobial activity against a wide range of microbes, including both Gram-positive (*Staphylococcus aureus*, *Bacillus subtilis*, *Streptococcus mutans*) and Gram-negative bacteria (*Escherichia coli*, *Pseudomonas aeruginosa*, *Salmonella enteritidis*, *Porphyromonas gingivalis*). For example, CNDs can inhibit the effect of *Phytophthora infestans* bacterial and fungal plant pathogens by silencing the genes of their dsRNA [61]. Moreover, vitamin C-derived CNDs have been shown to be effective against *S. aureus*, *E. coli*, and *B. subtilis* [62]. Furthermore, Jhonsi et al. [63] reported that CNDs derived from tamarind can against *S. aureus*, *E. coli*, and *P. aeruginosa*. The precursors are not only from natural sources but also synthetic materials such as ciprofloxacin hydrochloride and metronidazole (Table 2). The properties of CNDs could be modified or completely changed with the change in precursors [60]. The synthesis of CNDs from antibiotics was applied to expand the antibacterial properties of CNDs. Hou et al. [64] found that using ciprofloxacin hydrochloride to synthesize CNDs can enhance the antibacterial properties of CNDs, which resulted in the inhibition of the growth of both *S. aureus* and *E. coli*. Metronidazole, another antibiotic, was also used to produce CNDs, which can be against *S. mutans*, *E. coli*, and *P. gingivalis* [65]. CNDs produced from D(1)-glucose monohydrate and diethylenetriamine exhibit specific antibacterial activity only against Staphylococcus species bacteria [66]. Although CNDs have the potential to be a powerful tool for controlling microbial populations in agroecosystems, more research is needed to determine the best ways to use CNDs in agriculture, but this is a promising area of research.

In addition to the CNDs, magnetosomes are magnetic nanoparticles coming from a natural source like magnetotactic bacteria and can be a suitable alternative to chemically synthesized nanoparticles. Among others, functional magnetic nanomaterials based on iron, iron oxide, cobalt, and nickel ferrite nanoparticles, etc., are currently being investigated in agricultural applications due to their unique and tunable magnetic properties, the existing versatility regarding their (bio)functionalization, and in some cases, their inherent ability to increase crop yield [67]. Cobalt nanoparticles treasure high magnetic anisotropy properties and can be easily magnetized in one direction. For this reason, cobalt nanoparticles can be exploited in applications like energy storage or used as antimicrobial agents based on the observed high antibacterial activity [68].

**Table 2 nanomaterials-14-01249-t002:** Antimicrobial activity of carbon nanodots (CNDs) synthesized from different sources.

Microbe Species	Applied CNDs Dose	Types of CNDs and Their Precursors	Refs.
Gram-positive pathogenic organisms			
*Staphylococcus aureus*	100 μg mL^−1^ for 48 h	Vitamin C-derived CNDs	[62]
	200 μg/mL for >48 h	CNDs from tamarind plant (*Tamarindus indica* L.)	[63]
	250 μg/mL for 60 min for spermidine-based CQDs	CNDs from three biogenic polyamines (i.e., spermidine, putrescine, and spermine)	[69]
	100 μg/mL for 24 h	CNDs from henna plant (*Lawsonia inermis* L.)	[70]
	75 μL mL^−1^ for 12 h	CNDs from 2,20-(ethylenedioxy)-bis(ethylamine) and malic acid	[71]
	100 μg mL^−1^ for 12 h	CNDs from ciprofloxacin hydrochloride	[64]
	256 μg mL^−1^ for 18 h	CNDs produced from D(1)-glucose monohydrate and diethylenetriamine	[66]
	(7.8–62.5) µg/L for 24 h	CNDs from olive solid wastes in a hybrid form of ZnO–CNDs	[72]
*Bacillus subtilis*	100 μg mL^−1^ for 48 h	Vitamin C-derived CNDs	[62]
*Streptococcus mutans*	300 μg mL^−1^ for 24 h	CNDs from metronidazole under hydrothermal process	[65]
Gram-negative pathogenic organisms			
*Escherichia coli*	100 μg mL^−1^ for 48 h	Vitamin C-derived CNDs	[62]
	200 μg/mL for >48 h	CNDs from tamarind plant (*Tamarindus indica* L.)	[63]
	250 μg/mL for 60 min for spermidine-based CQDs	CNDs from three biogenic polyamines (i.e., spermidine, putrescine, and spermine)	[69]
	100 μg/mL for 24 h	CNDs from henna plant (*Lawsonia inermis* L.)	[70]
	75 μL mL^−1^ for 12 h	CNDs from 2,20-(ethylenedioxy)-bis(ethylamine) and malic acid	[71]
	100 μg mL^−1^ for 12 h	CNDs from ciprofloxacin hydrochloride	[64]
	256 μg mL^−1^ for 18 h	CNDs produced from D(1)-glucose monohydrate and diethylenetriamine	[66]
*Pseudomonas aeruginosa*	200 μg/mL for >48 h	CNDs from tamarind plant (*Tamarindus indica* L.)	[63]
	250 μg/mL for 60 min for spermidine-based CQDs	CNDs from three biogenic polyamines (i.e., spermidine, putrescine, and spermine)	[69]
	200 μg/mL for >48 h	CNDs from aminoguanidine and citric acid	[73]
*Salmonella enteritidis*	250 μg/mL for 60 min for spermidine-based CQDs	CNDs from three biogenic polyamines (i.e., spermidine, putrescine, and spermine)	[69]
*Porphyromonas gingivalis*	300 μg mL^−1^ for 24 h	CNDs from metronidazole under hydrothermal process	[65]

#### 4.1.1. Microbial Interactions for Synthesis of CNDs

Table 3 shows several studies collected that have used microbes for the green synthesis of CNDs from organic materials. Fang et al. [74] demonstrated the potential of this approach by using *Bacillus subtilis* to produce red emission carbon dots from leaves. Similarly, Wang et al. [75] utilized *cyanobacteria* to synthesize water-soluble CNDS with low cytotoxicity and high photostability. These studies collectively underscore the potential of microbial-based green synthesis to produce CNDs with various applications.

#### 4.1.2. CNDs for Promoting Plant Growth and Stress Tolerance

CNDs have been shown to enhance plant growth and stress tolerance in various studies summarized in Table 4. Su et al. [85] found that CNDs significantly improved the stress resistance of peanut plants. Li et al. [2] demonstrated that CNDs derived from *Salvia miltiorrhiza* triggered ROS-independent Ca^2+^ mobilization in plant roots, enhancing environmental adaptability. Wang et al. [86] showed that soil application of CNDs improved nitrogen bioavailability, promoting the growth and nutritional quality of soybeans under drought stress. Kou [87] synthesized nitrogen and sulfur co-doped CNDs, which enhanced drought resistance in tomato and mung beans by promoting seed germination and seedling physiology under drought stress. These studies collectively suggest that CNDs have the potential to be used as a tool for enhancing plant growth and stress tolerance. Furthermore, CNDs were reported to have a positive effect on rice plants in both phases, including seed germination and plant growth [88,89].

#### 4.1.3. Phytotoxicity Concerns of CNDs

Research has shown that CNDs can have both positive and negative effects on plants, depending on their concentration and the specific type of CNDs (Table 5). At high concentrations, CNDs have been found to reduce gas exchange and photosynthesis rates in *Arabidopsis thaliana* [97]. Furthermore, CNDs have been found to reduce root and shoot growth in maize [98]. However, they can also alleviate the toxicity of heavy metals such as cadmium in plants, as seen in studies on *Citrus maxima* seedlings [99] and wheat seedlings [100]. The toxicity of CNDs can vary depending on their source, with biochar-derived CNDs from different plant materials showing different levels of ecotoxicity [98]. The toxicity of water-soluble CNDs to maize was found to be concentration-dependent, with high concentrations significantly reducing root and shoot fresh weight [98]. These findings suggest that while CNDs can have negative effects on plants at high concentrations, they also have the potential to mitigate the toxicity of heavy metals.

#### 4.1.4. The Transportation of CNDs in Plant Organs

CNDs are shown to be transported via the vascular system in plants, as demonstrated by the enrichment of pollen-CNDs in the periplasmic space of *Brassica parachinensis* L. [104]. This transportation has also been observed in maize, where water-soluble carbon nanodots were found in the root-cap cells, cortex, and vascular bundle of roots, as well as in the mesophyll cells of leaves [105]. The rapid distribution and excretion of biocompatible CNDs synthesized using *Punica granatum* L. peel have been demonstrated in mice, suggesting their potential for use in biological imaging and drug delivery [105]. These studies collectively indicate that CNDs can be transported within plant organs, with potential applications in plant imaging and nutrient tracking. The properties of CNDs, including their size, oxygen content, and ability to enhance photosynthesis, can influence their transport within plant organs. Wang et al. [106] and Li et al. [88] both found that CNDs can promote plant growth and photosynthesis, with Li et al. [88] specifically noting their ability to penetrate all parts of rice plants. Li et al. [107] added that CNDs can be transferred from roots to stems and leaves through the vascular system, suggesting their potential as delivery vehicles in plants. These studies collectively suggest that the properties of CNDs can influence their transport within plant organs, potentially leading to enhanced growth and photosynthesis.

### 4.2. Nano-Tellurium: A New Frontier in Nano-Microbe Nexus

#### 4.2.1. Why Nano-Tellurium?

Tellurium is an element of the chalcogen group, including oxygen, sulfur, selenium, and polonium. In 1782, Franz-Joseph Mueller von Reichenstein (1742–1825), conducting research on gold-containing ores, made a relatively early historical discovery. Originating from the Latin word “Tellus”, which means “Earth”, the phrase tellurium means a non-essential biological metalloid element that is a member of the chalcogen family is tellurium (Te) [108]. In contrast to selenium, tellurium’s biological and therapeutic characteristics, and its derivatives have not been thoroughly studied [109]. Out of all the Te species, such as tellurides (Te^2−^ [+2]), tellurite (TeO_3_^2−^ [+4]), and tellurate (TeO_4_^2−^ [+6]), only elemental tellurium (Te^0^) is water-insoluble which can be sorted at the nanoscale through chemical or biological reduction [110]. Microbes can be highly toxic to both Te oxyanions, with TeO_3_^2−^ [+4] being more lethal than TeO_4_^2−^ [+6] [111]. Above all, it is noteworthy that toxicity to pathogens is also accompanied by toxicity to the host. Therefore, the environment, as well as the flora and fauna, may benefit from the synthesis of tellurium nanoparticles (Te-NPs) with adjustable features. Additionally, employing biomaterial with the Te-NPs might improve their biological characteristics while having minimal toxicity [112]. Table 6 consists of a few examples of the antimicrobial activities of Te-NPs. The promising applications of nano-tellurium are presented in more detail by [113].

#### 4.2.2. Antimicrobial Mechanisms of Nano-Tellurium

Metal nanoparticle’s antibacterial properties result from their capacity to damage membranes of cells, block enzymes, trigger the production of reactive oxygen species, and restrict microbe entry to essential trace elements. Certain metals may also be directly toxic to DNA [130,131,132,133]. The steps belong the antimicrobial mechanisms of nano-Te can be noticed in Figure 1. This figure illustrates the different antimicrobial mechanisms of metal nanoparticles and their reactions with pathogens’ intra- and extracellular components. These steps may include producing reactive oxygen species (ROS), release of ions, inhibition of biofilm formation, removal of biofilm, and interaction with membranes.

#### 4.2.3. Nano-Tellurium: Microbial Synthesis

Several studies reported on the microbial or biosynthesis of nano-tellurium such as bacteria of *Lysinibacillus* sp. EBL303 [134], *Aromatoleum* sp. CIB [126], *Streptomyces graminisoli* [122], as well as fungi of *Mortierella* sp. AB1 [127], *Aspergillus niger* [135], *Aureobasidium pullulans* [136] and Archaea of *Haloferax alexandrinus* GUSF-1 [137]. This biosynthesis of Te-NPs can be usually formed in the periplasmic space, cytoplasm, or outside the cell. Producing biological Te-NPs includes both reduction and precipitation, which may be related to reducing substances or the reductase at different sites occurring either extracellularly or intracellularly [138]. The plant-associated microbes can play a crucial role in improving plant health under environmental stresses, especially in the presence of applied nanomaterials [139]. There is little evidence available for the transport mechanism of nano-tellurium in plant and soil systems. Since Te-NPs have huge potential as antimicrobial agents, they should be examined against multiple plant pathogens and their mode of action [140]. It has been reported that the plants can remediate the Te in the soil. Nano-sensing can be employed to develop a method for the rapid estimation of the Te-NPs and other Te species. On the other hand, the nexus of nano-Te–plant–microbe still needs to be investigated on different levels or conditions.

## 5. Transport of Nanomaterials to Different Organs

Understanding the transport of nanoparticles into plants is crucial for harnessing their potential in agriculture and biotechnology. The journey of nanoparticles from external environments to internal plant tissues involves intricate processes influenced by both the properties of the nanoparticles and the characteristics of plant structures. Nanoparticles can be designed to transport specific materials, compounds, or genetic information within plants, acting as carriers to deliver these payloads to target locations. The controlled delivery of nanoparticles can be advantageous for precise and efficient plant applications [141]. Several environmental factors such as soil pH, temperature, oxygen gradient, and relative humidity play pivotal roles in determining the fate and impact of nanoparticles [142]. Moreover, the size and shape of nanoparticles can influence their ability to enter plant cells or tissues. In some cases, specific shapes or sizes may be more effective [143]. Different plant species or crops may have varying requirements and responses to nanoparticle-based applications, affecting the selection of nanoparticle types for their transportation to specific plant organs [144].

### 5.1. Soil Application and Uptake of NPs

NPs undergo a range of bio/geo-transformations in the soil that affect their harmful effects and bioavailability. NPs travel to apical regions and congregate in subcellular structures or cells after connecting with root networks. The first stage of biological accumulation is the absorption of NPs from the soil through the roots of plants [145,146]. Moreover, it has been noted that microscopic NPs (dimensions spanning 3 to 5 nm) can enter root systems simply via root epidermal cells or by capillary forces [147,148]. The primary epidermal cells construct a semipermeable cell wall with tiny pores that effectively restrict the large-size NPs. Initial pore formation in the epidermal cell wall simplified the uptake of some NPs [149,150]. After passing through extracellular spaces and through cell membranes, NPs reach the central vascular cylinder, which permits the xylem to rise vertically. NPs must traverse the Casparian strip barrier via a symplastic pathway to enter the core vascular cylinder (Figure 2). This happens through endocytosis, pore establishment, and delivery after attaching to carrier proteins in the membrane of endodermal cells. NPs, embedded in the cytoplasm, can move across cells via plasmodesmata [151,152]. The Casparian barrier gathers the NPs that cannot get inside the plant, while the shoots and roots accept the NPs that have reached the xylem [153]. Absorbed NPs may be placed in the outer layer cell membrane, cortical cell inner environment, or centers of plants. Conversely, non-absorbed NPs on the root surface of a soil aggregate can change nutrient absorption [152,153]. When laying the seed on the soil, seeds can take in the soil-blended NPs instantly through the coat using parenchymatic intercellular spaces, with consequent NP diffusion in the cotyledon [152,154,155].

### 5.2. Foliar Application and Uptake of NPs

In agriculture, foliar sprays of engineered NPs are used more and more as nano-fertilizers, nano-pesticides, nano-sensors, and nanocarriers. When matched to the standard process of soil–root treatment, the efficacy of plant protection technologies is enhanced when NPs are sprayed directly onto the leaves. Spraying foliar solutions with NPs allows them to go into the plant primarily through the stomata and then travel throughout the plant via apoplastic and symplastic pathways (Figure 2). The practice of foliar NPs has been revealed to increase crop yield and quality, as well as increase plant defenses against pests and diseases. The processes by which foliar NPs trigger harm, however, remain to be fully elucidated. In addition, the chemical and physical features of NPs and abiotic factors like temperature, humidity, and light should be explored to better this technology’s ability to enhance foliage uptake of NPs. The NPs that are put on the leaves can get in through the stomata or cuticles [156]. The cuticle is the first line of defense for a leaf, preventing particles smaller than 5 nm from entering the plant. They get into the plant through stomata, and their cells place them in the plant’s vascular system through apoplastic and symplastic pathways [157]. NPs that are between 10 and 50 nm tend to move through the cytoplasm of the cell next to them (symplastic route). Thus, NPs ranging in size from 50 to 200 nm migrate between cells (apoplastic pathway).

Adopted NPs move through the sugar-solution-filled phloem sieve tubes. The roots, stems, fruits, grains, and young leaves all act as powerful sinks for the sap, so NPs can move in both directions as they are transported via the phloem in the plant’s vascular system [158]. As a nonselective path of least resistance, the apoplastic pathway is well-known. It is commonly accepted that the apoplastic route is the most efficient for the translocation of numerous water nutrients and non-essential metal complexes [159]. Applicable adsorption of NPs following foliar application was determined by application method, NP size and concentration, and environmental conditions [160]. Many factors, including the morphology of leaves and chemical composition, trichrome presence, and the presence of leaf exudates and waxes, influence the ambushing of NPs on the leaf surface [150].

The trichomes on plant organs can modify surface activity by capturing NP on the surface of the plant, lengthening the time that exogenous materials remain on tissues [161]. Impairments and wounds in plants above ground and hypogeal parts can be effective pathways to internalize NPs. Trapping of NPs on the leaf surface is impacted by numerous leaf morphological elements, including leaf form and chemical make-up, the existence of trichrome, leaf exudates, and waxes [162].

### 5.3. Translocation of NPs

The plant’s body translocation is divided into two important parts: the apoplast and the symplast. Apoplast-based translocation of nutrients occurs via interconnected cellular membranes found on the inside of plant cells. On the other hand, symplast-based translocation of nutrients occurs via the protoplasts of different plant cells that are connected by a thin cytoplasmic connection. These are two routes through which dissolved ions can get into and out of the roots. Ions can only enter root cells through symplastic pathways via membrane-specific channels and transporters that allow them to cross the plasma membrane (Figure 2). Apoplastic movement has been demonstrated to promote NPs’ circular motion, which may carry NPs to the vascular tissues and the root’s core cylinder, allowing their ascent into the plant’s apical portion. This method of NP translocation is particularly useful for those uses that call for systemic NP delivery. However, a layer of lignin-like structures called the Casparian strip prevents the root endodermis from completing its radial migration. NPs must eventually enter cells to undergo symplastic transport. Plant cells are more challenging than animal cells to deliver NPs intracellularly since they have a strong cell wall that functions as an external barrier to the cell entrance. Different cell entrance mechanisms have been identified in cells, including those that rely on the formation of holes, membrane translocation, or carrier proteins [163]. Plasmodesmata, which are cytoplasmic bridges (membrane-bound) having an adjustable diameter (20–50 nm), aid in the migration of NPs from one cell to another after they penetrate the cytoplasm. Research has been conducted on Arabidopsis, rice, and poplar to characterize the movement of different-sized NPs via plasmodesmata [151]. Small particles can translocate throughout the entire plant via the symplastic and apoplastic passageways, making their way to the xylem and phloem vessels. Interestingly, NPs tend to accumulate in organs that are highly capable of pulling in phloem fluids (sink activity), such as flowers, fruits, and seeds. Concerns about NP buildup in particular organs are equally as chief as concerns about NP toxicity to plants [151].

## 6. Nanotoxicity on Soil Microbes and Plants

### 6.1. Toxicity Concerns about Soil Microbes

NPs have interactions with environmental biological systems, changing surface characteristics and affecting microbial communities. They may alter the accessibility of toxins or nutrients, increase the harmful effects of permanent organic contaminants, or have direct or indirect toxic effects. Possible mechanisms include damage to cell membranes, protein oxidation, genotoxicity, interaction with respiratory chains, and reactive oxygen species production or apoptosis [164,165]. According to the previous discussion, most of the research that has been published on the topic of ENMs and NPs toxicity to microorganisms has focused on mechanisms occurring at the cellular level, which explains how these microbes function. For example, NPs interact differently and cause varying degrees of toxicity with Gram-positive and Gram-negative bacteria due to differences in the cell wall composition of these species, which include phospholipid bilayer, lipopolysaccharides, and peptidoglycans (Figure 3). The results of more recent evaluations corroborate this. The bacterial cell wall composition and charges are major factors in the adherence of graphene NPs and the resulting toxicity [166,167].

It has been discovered that NPs, such as TiO_2_, ZnO, CuO, Ag, carbon nanotubes (CNTs), and fullerenes, are extremely dangerous to beneficial plant microbes engaged in the nitrogen cycle, organic carbon breakdown in soil, and nutrient absorption (Figure 3) [166,167]. These microbes have the ability to diminish soil microbial ecosystems and consortiums that support plant development, like mycorrhiza and rhizobacteria. Metal ion poisoning can occasionally result from these NPs dissolving in soil solutions or salt water. Using a pure culture medium, research has been conducted on the toxicological impacts of Ag-NPs on different types of bacteria. These bacteria include *Escherichia coli*, *Pseudomonas aeruginosa*, *Bacillus subtilis*, *Staphylococcus aureus*, *Nitrobacter*, and *Nitromonas* [168,169,170]. For instance, Beddow et al. [171] examined the effects of Ag_2_SO_4_ and covered and uncovered nano-Ag on the functioning of several bacterial species. The nitrification ability rates and proliferation of the microbes, comprising *Nitrosomonas europaea*, *Nitrosospira multiformis*, *Nitrosococcus oceani*, and *E. coli*, were shown to be drastically lowered by all treatments employing Ag-NPs. Results showed that capped nano-Ag had the greatest inhibitory effect on bacterial growth, followed by uncapped nano-Ag and Ag_2_SO_4_. It can be difficult to generalize results from these types of investigations conducted in controlled lab settings using pure microbial cultures to real-world settings. Environmental factors, including soil and sediment systems, can act as a sink for NPs, either decreasing or increasing their bioavailability, depending on the exposure. Validating the toxicological assessment in the adsorbent system, like soils and sediments, would be the most appropriate course of action [172]. Similarly, CNMs can have a direct harmful effect on soil microbes, affect the toxicity of organic substances in the soil, or interfere with the bioavailability of nutrients. When NPs are harmful to plants, they may also have an indirect effect on symbiotic microbes [173]. For example, it has been discovered that C60 can limit the growth of bacteria that are often present in soil and water. Highly effective oxidizing agents in biological organisms can be hydroxylated C60 or C60-coated polyvinyl pyrrolidone ENMs, as they generate ^1^O_2_, which can cause lipid peroxidation and cell destruction [174]. Studying the effects of C60 and CdSe quantum dots (QD) on nitrate-reducing bacteria-mediated organic matter oxidation in freshwater sediments, researchers discovered that C60, at 140 µg/L, completely inhibited microbial acetate oxidation, while CdSe QD, at 200 µg/L, slowed the rate of acetate oxidation in the sediment slurries [175]. According to some research, the main reason why QDs are poisonous to bacteria is because they release toxic components, including ions or heavy metals, that the bacteria already have in their core or shells [175]. Quantum dots have powerful harmful effects on microbial communities, yet there are few extensive reports on their stability and dissociation in the environment. Similarly, CNMs can have a direct harmful effect on soil microbes, affect the toxicity of organic substances in the soil, or interfere with the bioavailability of nutrients. When ENMs are harmful to plants, they may also have an indirect effect on symbiotic microbes [173]. For example, it has been discovered that C60 can limit the growth of bacteria that are often present in soil and water. Because they produce singlet oxygen, which can induce lipid peroxidation and cell damage, hydroxylated C60- or C60-coated polyvinyl pyrrolidone ENMs can function as powerful oxidizing agents in biological systems [174]. Research on the impact of C60 and CdSe quantum dots (QD) on organic matter oxidation in freshwater sediments by nitrate-reducing bacteria found that C60 at a concentration of 140 µg per liter entirely blocked the microbial oxidation of acetate, while CdSe QD at a concentration of 200 µg per liter retarded the rate of acetate oxidation in the sediment slurries [175]. According to some research, the main reason why QDs are poisonous to bacteria is because they release toxic components, including ions or heavy metals, that the bacteria already have in their core or shells [175]. Quantum dots have powerful harmful effects on microbial communities, yet there are few extensive reports on their stability and dissociation in the environment. NPs made of metal oxides are poisonous to soil microbes. The most prevalent and extensively utilized metal oxide NPs in many goods are those based on Zn, Cu, and titanium, and there are a plethora of studies and evaluations addressing their toxicity. The photocatalytic activation of TiO_2_ often inhibits bacterial development by forming ROS and H_2_O_2_ because of UV exposure, which in turn causes cell death. Metal oxide-based ENMs in nature are likewise affected by the same kinds of environmental factors. Soil organic matter quantity and form, which enhance ENMs’ propensity to aggregate and interact with biomolecules, and soil pH, which affects ENMs’ effective cation exchange capacity, are environmental toxicity-governing factors [176]. But it is also true that soil has varying effects on various orders of bacteria [177].

### 6.2. Toxicity Concerns in Plants

#### 6.2.1. Morpho-Physiological Concerns

A frequent harmful consequence of metal-based NPs is the suppression of seed germination (Figure 4). Possibly owing to the dissolved Zn cation toxicity, Lin and Xing (2008) discovered that Zn NPs on ryegrass and ZnO-NPs on maize inhibited seed germination at 2000 mg L^−1^ [147]. El-Temsah and Joner (2012) investigated the harmful impact of zero-valent Fe-NPs and Ag-NPs, having particle sizes ranging from 1 to 20 nm, on the germination of ryegrass, barley, and flax seeds (Figure 4) [178]. The concentrations tested ranged from 0 to 5000 mg L^−1^ of zero-valent ion NPs and from 0 to 100 mg L^−1^ of Ag, respectively. The inhibitory effects of zero-valent iron NPs in aqueous solutions were seen in their investigation at 250 mg L^−1^. At concentrations between 1000 and 2000 mg L^−1^, zero-valent iron NPs were found to completely suppress germination. At lower doses, Ag NPs decreased seed germination; however, these effects were not size-dependent and were never entirely attenuated.

The use of NPs has both a negative and beneficial impact on seed germination. According to the investigation by Feizi et al., TiO_2_ NPs at concentrations of 2 and 10 mg L^−1^ have been proven to accelerate wheat germination and boost biomass [179]. There appears to be a concentration dependency for the impacts on many germination parameters, with either a null effect, a beneficial effect occurring at low concentrations, or an inhibitory effect occurring at a high Zn NPs concentration. Multiple crops, including *Zea mays* [180,181], *Solanum lycopersicum* [182,183], *Lathyrus sativus*, *Raphanus sativus* [184,185], and *Sinapis alba* [186] have been studied. Depending on the study, germination can be either inhibited or unaffected, as seen in the cases of *L. sativa* and *R. sativus* [184,187], *S. lycopersicum* [188], and *Oryza sativa* (Figure 4) [189]. The effect on wheat germination varies with the degree of oxidation of the Cu in the NPs. Low quantities of Cu^2+^ were shown to promote germination, but Cu^1+^ had little impact [190]. Above a specific dose (0.25–0.5 mg L^−1^), depending upon the targeted crop, detrimental effects on germination are seen with Fe NPs (Figure 4) [178]. Root elongation in all the species studied came to a near halt when they were subjected to Zn NPs or ZnO NPs in suspensions of 2000 mg L^−1^. Zn NPs and ZnO NPs were predicted to have 50% inhibitory concentrations (IC50) of approximately 50 mg L^−1^ for radish and around 20 mg L^−1^ for rapeseed and ryegrass. The effects of phenanthrene (Phen) on the growth of maize, cucumber, soybean, cabbage, and carrot roots [191]. Particle-induced suppression of root growth was shown to be attenuated when particles were loaded with either 10.0%, 100.0%, or 432.4% monomolecular layer (MML) of Phen. In their research, Ma et al. studied the susceptibility of seven different plant species—including radish, rape, tomato, lettuce, wheat, cabbage, and cucumber—to four unusual metal oxide NPs throughout the stages of root elongation: CeO_2_ NPs, La_2_O_3_ NPs, Gd_2_O_3_ NPs, and Yb_2_O_3_ NPs (Figure 4) [192]. Root development was significantly affected by NPs lying in a wide range. At a concentration of 500 mg L^−1^, ZnO NPs are found to cause oxidative stress inside soybean (*G. max* L.) seedlings. Root cell survival, root stiffness, and plant development were all significantly reduced by ZnO NP stress. The oxidation–reduction cascade-related genes were downregulated after being treated with ZnO NPs. These genes included GDSL motif lipase 5, SKU5 similar 4, galactose oxidase, and quinone reductase [193]. Dimkpa et al. (2015) evaluated the influence of commercial ZnO (<100 nm) NPs on wheat (*T. aestivum* L.) grown in a solid matrix or sand. ZnO NPs acted similarly to their bulk counterparts in solubilizing metals in the sand. ZnO NPs (500 mg kg^−1^) added to the sand considerably (*p* = 0.05) inhibited root development; however, this effect was mitigated by using the bulk amendment [194]. Liu et al. (2016) investigated how aquatic medium-germinated lettuce (*Lactuca sativa*) seeds were affected by laboratory-prepared FeOx NPs (likely γ-Fe_2_O_3_) [195]. Root elongation was increased by 12–26% over a concentration range of NPs (5–20 mg L^−1^), and they were much less harmful than their ionic counterparts. In contrast, at 50 mg L^−1^, root growth was suppressed.

#### 6.2.2. Cellular and Biochemical Concerns

Some NPs damage cells and subcellular organelles, causing cell membrane damage and mitochondrial dysfunction (Figure 4) [196]. When compared to the control group, the treatment with Al_2_O_3_ NPs changed the structure of the soybean cell walls, resulting in minute fissures close to the tips of the roots and damage to the root cap (Figure 4) [197]. After being absorbed by soybeans, Al_2_O_3_-NPs promoted ROS generation in mitochondria and chloroplasts, suggesting that ROS may contribute to cell injury [198]. In a study on *Vigna radiata* and *Brassica campestris*, researchers found that Ag NPs might enter cells and damage vacuoles as well as cell walls, potentially impacting other organelles [199]. Tripathi et al. discovered that Ag and ZnO NPs reduced vacuole size and cell turgidity in maize and *B. oleracea* [182]. TEM pictures showed PS-NPs breaking chloroplast structure and damaging lettuce cells [189]. FTIR as well as synchrotron computer micro-tomography demonstrate that PS-NPs (Pd-doped PS-NPs) may impact wheat root cell walls, modifying root anatomy [200]. Chemically manufactured CuO NPs were more harmful than biologically created ones in physiological assessments [201].

The importance of structural modification in NP toxicology is demonstrated by research on the cytotoxic impacts of SnO_2_ and Ag/SnO_2_ NPs in tobacco cell cultures. In tobacco cells, SnO_2_ NPs demonstrated low toxicity, whereas Ag-doping caused toxicity through oxidative stress. Microscopic analysis revealed cell death in high-level SnO_2_ NP treatments (0.5 mg/mL) and a high NR concentration in tobacco cells exposed to NP stress, indicating vacuolar pH acidification (Figure 4) [202]. In *O. sativa* L., callogenesis and rejuvenation were enhanced by biosynthesized CuO NPs (1–20 mg/L) derived from Azadirachta indica leaf extract. The highest regeneration rates were seen in Basmati 385 (92%) at 20 mg L^−1^, followed by Basmati 2000 (80%), Super Basmati (52%), and Basmati 370 (32%). Low CuO-NP levels (1 and 15 mg/L) resulted in poor Basmati 370 regeneration [203]. In order to stimulate cellular responses to changes in their environment, plants, like other aerobic organisms, choose ROS as a signal molecule [204,205,206]. When NPs reach plant cells, they can alter ROS levels, impede cell metabolism, damage the antioxidant system, and ultimately stunt plant development. OH, ^1^O_2_, O_2_^−^, and H_2_O_2_ are all examples of ROS that are created continually and naturally by cell organelles during metabolism [207,208,209]. Both low levels of ROS (which activate a defensive signal) and high levels of ROS (which cause oxidative damage to amino acids, lipids, and nucleic acids) are crucial [210]. The level of lipid peroxidation in a cell’s membrane is a useful indicator of membrane health [211,212,213]. Lipid peroxidation caused by ROS production causes membrane damage, which in turn causes ion leakage, metabolic disruption, and cell death. Cells and subcellular components are protected against the harmful effects of active O_2_∙ by the antioxidant enzymes and low-molecular-weight antioxidants found in plants [214]. As a result, most of the research on the oxidative damage caused by NPs to plants concentrates on measuring ROS, or antioxidant levels, as well as antioxidant enzyme activity. ROS were produced in response to Ag-NPs on bean leaves. Since smaller NPs have a greater specific surface area and provide more cytotoxicity, the findings showed that ROS generation increased with decreasing particle diameters [215]. The presence of TiO_2_ NPs under intense sunlight might cause a rise in tocopherol levels and a reduction in CAT action in SOD activity. Higher quantities of NPs promote membrane lipid peroxidation, which may be related to particle photoactivation and increased ROS generation [216]. Although lettuce’s antioxidant enzyme activity rose in response to the oxidative stress brought on by polymethyl methacrylate nanoplastics (PMMA-NPs), this did not prevent further damage from free radicals. Still, it had a higher concentration of active oxygen than the control [217]. Researchers have recently proven that plants are able to boost defense responses and reduce excessive accumulation of ROS when exposed to increasing concentrations of polymeric NPs in their root systems [218]. Electron leakage to O_2_ at the time of electron transport in chloroplasts, mitochondria, and plasma membranes is a common source of ROS [219]. When ROS builds up to dangerous levels in plant cells, equilibrium is lost. When the quantity of ROS surpasses the capacity of the defense system, the cells undergo a condition known as “oxidative stress”, which causes irreversible damage to the lipids, proteins, and nucleic acids found inside the cell membranes as well as the stimulation of the production of associated defense genes in reaction to such damages [211].

In response to NP stimulation, plants release ROS, which has several functions, including removal [66,220,221,222,223]. However, ROS in excess degrades biomolecules and ultimately kills cells. Eliminating ROS is the job of antioxidant enzymes [224]. Tarrahi et al. (2017) evaluated the effect of Se NPs capped with L-cysteine and tannic acid on *L. minor* by measuring changes in antioxidant enzyme activity; they found that Se NPs had no effect on superoxide dismutase (SOD) activity but had a significant depressing effect on peroxidase (POD) activity. CAT activity rose as opposed to that of POD [225]. To further assess the toxicity of ionic and NP Se, sodium selenite (Se^4+^) has been studied. Moreover, the ionic forms of Se increased the activities of SOD and CAT while significantly inhibiting POD. In addition, ions had a suppressing effect on POD due to the ROS burst’s ability to denature the enzyme’s structure. For SOD, it is determined that low Se^4+^ concentrations, as well as Se NPs, could not appropriately disrupt the defense system response for POD, but it is probable that the molecular structure has been altered [225,226].

The uncontrolled ROS formation that occurs during abiotic stress can be neutralized by a combination of nonenzymatic substances such as glutathione, suitable solutes, phenolics, alpha-tocopherol (vitamin E), carotenoids, flavonoids, and proline [227]. Changes in flavonoids and phenols are common when environmental factors, including UV-B radiation, dryness, and heavy metals, are present. Nonantioxidant enzymes can get rid of ROS [228,229,230,231]. In three separate studies, it was found that L-cysteine and tannic acid-capped Se NPs, ZnSe, and CdSe NPs increased the phenol and flavonoid contents of *L. minor* [225,232,233]. Ag NP and ZnO NPs had a similar effect on castor seedlings and *Brassica nigra* [234,235]. This growth is associated with ROS scavenging and chelation. The buildup of MDA could be used to probe membrane integrity [236].

#### 6.2.3. Molecular Concerns

When NPs are liberated into the environment, they tend to accumulate in different food tiers. Plants can absorb NPs, but they are very vulnerable to nanotoxicity (Figure 4) [237]. That is why plants are held in such high esteem as genetic models for screening and keeping tabs on potentially harmful substances in the environment [182]. Because NPs can interact with cellular macromolecules such as the nucleus, cellular elements, or lipids, they may have cytotoxic and genetically toxic effects on plants. These effects include a hike in the chromosomal abnormality index and a reduction in the mitotic index, respectively [238]. Cyclins, cyclin-dependent kinases (CDKs), and checkpoint kinases are only a few of the many proteins and enzymes that play a role in controlling the cell cycle. The cyclin protein family binds to cyclin-dependent kinases (CDKs) and activates them, hence promoting cell cycle advancement. Proteins used in DNA replication, chromosomal segregation, and cell division are only some of the many targets for phosphorylation by CDKs. Checkpoint kinases detect blunders in the cell cycle and, if necessary, activate DNA repair or apoptosis pathways. Cell cycle checkpoints are DNA surveillance mechanisms that halt the cell division process at critical points to avoid the development of genetic mistakes. When irreparable DNA damage is detected, checkpoints can either halt the cell cycle’s progression or force the cell to exit the cell cycle or die [239].

Damage to cells caused by PS-NH_2_ NPs can result in a progressive, combined stoppage of the cell cycle between the G1/S and G2/M stages. It is interesting to note that despite the cell cycle stop, neither intracellular ATP levels nor NP internalization decreased [240]. Epigenetics is “the study of mitotically and/or meiotically heritable changes in gene function that cannot be explained by changes in DNA sequence” [241]. DNA methylation, histone modifications, and non-coding RNAs are just a few of the mechanisms that have been discovered during the past several decades to modulate DNA expressions without affecting the sequence itself [242]. The epigenetic modifications to plant DNA caused by nanomaterials have only recently begun to be studied. The purpose of this research was to ascertain if wheat cultivated in a medium containing NPs exhibited epigenetic modifications to its DNA [243]. Previous research has shown that ZnO NPs also alter the expression level of the HSFA4A gene in wheat by acting on transcription factors [244]. Zn may also function through its interactions with biomolecules and cellular organelles, which might explain the epigenetic polymorphism in callus tissues caused by ZnO-NPs [245]. Zn is an essential mineral since it is used in the production of chlorophyll, carbohydrates, and phytohormones [246,247]. It has been demonstrated that *Cyamopsis tetragonoloba* L., *Gossypium hirsutum*, *Lycopercicum esculentum*, and Stevia all benefit from Zn NPs. According to the literature (Ul Haq, 2019), plants grown with Zn NPs had longer shoots and roots, higher chlorophyll and protein content, and higher yields [248]. Increased generation of ROS may occur when an excessive number of NPs is present [249]. It has been found that metal NPs cause plants to experience stress. According to (Ul Haq, 2019), *Cyamopsis tetragonoloba* L. plants exposed to Zn NPs had considerable changes in biomass, lengths of the roots and shoots, amounts of protein and chlorophyll, and the activity of enzymes [248]. Abnormalities in the cell cycle and a decrease in biomass, production, and quality might be caused by micronutrient excess or deficiency, respectively [250,251].

Cu is essential for the metabolism of cell walls, transmission of electrons, and ethylene receptor facilitation [252]. Since Cu participates in many different physiological processes in plants, it finds most of its applications in agriculture [252]. Cu ions, for instance, prevent oxidative stress in plant cells [253], promote the formation of hydroxyl radicals [254], and aid in the activation of metabolic pathways [255]. At sublethal quantities, CuO NPs can disrupt the Krebs cycle, making them more soluble and poisonous [256]. CuO NPs were found to have a detrimental effect on lettuce, mung beans, kidney beans, alfalfa, wheat, chickpeas, and many other crops’ seed development and growth [257]. Changes in gene function that are passed down through generations but do not involve a change in the DNA sequence are included in epigenetics [258]. Histone modifications and DNA methylation are two instances of epigenetic events [206]. When compared to the average polymorphism percentage for MspI digestion, the higher polymorphism percentages observed in the present study following treatment with non-Fe_3_O_4_ NPs, as well as 2X ZnO and CuO, could be classified as hypermethylation. More and more studies are looking at the potential impacts of NPs on histone modifications as we learn more about NP-mediated changes in DNA methylation. The nuclear proteins known as histones have octameric structures with the subunits H2A, H2B, H3, and H4 duplicated twice. Histones facilitate the compact packaging of DNA in the nucleus by providing a scaffold for DNA to wind around and form a nucleosome [259,260].

Understanding the mechanism of nanoparticle toxicity will inform efforts to redesign nanoparticles with reduced environmental impact. The redesign strategies will need to be chosen based on the major mode of toxicity, but also considering what changes can be made to the nanomaterial without impacting its ability to perform in its intended application. To reduce interactions with the cell surface, nanomaterials can be designed to have a negative surface charge, use ligands such as polyethylene glycol that reduce protein binding, or have a morphology that discourages binding with a cell surface. To reduce the nanoparticle dissolution to toxic ions, the toxic species can be replaced with less toxic elements that have similar properties, the nanoparticle can be capped with a shell material, and the morphology of the nanoparticle can be chosen to minimize surface area and thus minimize dissolution, or a chelating agent can be co-introduced or functionalized onto the nanomaterial’s surface. To reduce the production of reactive oxygen species, the band gap of the material can be tuned either by using different elements or by doping, a shell layer can be added to inhibit direct contact with the core or antioxidant molecules can be tethered to the nanoparticle surface. When redesigning nanoparticles, it will be important to test that the redesign strategy reduces toxicity to organisms from relevant environmental compartments. It is also necessary to confirm that the nanomaterial still demonstrates the critical physicochemical properties that inspired its inclusion in a product or device [261].

## 7. Regulations of Nanotoxicity

### 7.1. Current Challenges of Nanotoxicity

#### 7.1.1. Risk and Challenges in Ecological Level

Soil microbes are infected by NPs made of metal oxides. The toxicity of metal oxide NPs based on Zn, Cu, and titanium, the three most frequent and extensively utilized metal oxide NPs in a broad range of products, has been the subject of several studies and evaluations (Figure 5) [262,263]. The usage of dysprosium oxide NPs (nDy_2_O_3_) and other lanthanide oxide-based NPs (LnO-NPs) is on the rise in the biomedical industries, but these particles have negative effects on natural biological systems and disrupt the metabolism and structure of bacteria like *E. coli* [264]. Using soil samples, Rousk et al. (2012) investigated the ecotoxicity of NPs based on Zn oxide and Cu oxide on a consortium of soil microorganisms [176]. The study utilized Zn- and Cu oxide NPs as well as two reference compounds: non-nanoparticulate bulk oxide and extensively soluble metal sulfate forms. The purpose of these was to determine whether the observed toxicity was caused by the nanoparticulate form of the metal or by the solubilization of metal ions in the soil solution. In addition, the study found that bulk (macro-particulate) CuO was non-toxic to soil bacteria, whereas CuSO_4_ and its oxide forms were far more harmful. Soil bacteria were killed off by all types of Zn; however, bulk ZnO was more hazardous than nano ZnO [265,266]. Additionally, the investigation showed a strong correlation between the growth of bacteria and the quantity of dissolved metals in the solution. The investigators concluded that the transformation of metal oxides and sulfides into their dangerous metal ion forms was the primary source of toxic effects. A comparative investigation of the environmental toxicity of TiO_2_, SiO_2_, and ZnO NPs to *E. coli* and *Bacillus subtilis* revealed that antimicrobial activity usually increased from SiO_2_ to TiO_2_ to ZnO and that *B. subtilis* was particularly vulnerable to such detrimental effects [267]. The in vivo and in vitro effects of NPs on microbes may also vary. Recent research on the impacts of TiO_2_ NPs (ranging in size from 10 to 100 nm) on *Drosophila* intestine commensal bacteria found that they could limit the growth of these bacteria in vitro in a way that was dose- and particle-size-dependent [264]. In addition to this, the in vivo gut microbes of *Drosophila* were unaffected by the identical dosage and particle size of TiO_2_. Another unexpected finding was that the inhibition did not depend on photocatalytic activation of TiO_2_, according to the study. When exposed to ultraviolet light, TiO_2_ typically undergoes photocatalytic activation, resulting in the production of ROS and H_2_O_2_, which in turn kill microorganisms. Similarly, ENMs based on metal oxides are affected by environmental factors. Several environmental factors govern ENM toxicity, including soil organic matter quantity and shape (which enhance ENM aggregate formation and biomolecule interaction) and ENM effective cation exchange capacity (which is affected by soil pH) [176]. On the other hand, many bacterial orders do experience soil effects in distinct ways [268,269]. Since NPs have been used more often over the past ten years, they have been released into aquatic habitats, endangering both plants and animals with serious toxicity. AgNPs harmful effects are influenced by organisms, chemistry, solubility, and bioavailability. Research has indicated adverse impacts on the development of plants, disturbance of algal growth cycles, and possible damage to aquatic vegetation. In nanomedicine, NPs are also employed as carriers for medications. However, when discarded, they can end up in aquatic habitats as soluble ions and aggregated particles, which can be extremely harmful to marine creatures.

While certain NPs might not be intrinsically dangerous, their interaction with common trash might contaminate harmful substances, leading to their absorption and easier entry into cells. These creatures are at risk due to the rising concentration of NPs in marine environments.

#### 7.1.2. Human Health-Related Risks and Challenges

NPs have the potential to be harmful through several methods, such as invasion into the central nervous system and interactions with biological fluids like blood and tissues. Physical interaction, oral intake, or inhalation are the three ways that exposure can happen. Breathing, direct skin or eye contact, and absorption via the digestive tract are all common ways. Certain NPs could be hydrophobic, which would cause them to build up in the liver and spleen. Soluble metals or elements liberated from the particles might have detrimental effects on biological systems, which is frequently the cause of the adverse consequences. Chemically generated NPs are more hazardous to human cells since they include harmful compounds and agents.

On the other hand, biosynthesized NPs have surface functional groups that are compatible with living organisms, which can lessen their toxicity [270]. CNTs are one type of NP that can mimic the effects of asbestos exposure on cells while being much smaller than asbestos [271]. Metal NPs (e.g., SiO_2_ NPs, ZnO NPs, TiO_2_ NPs, Ag-NPs, and Au NPs) have been connected to several detrimental impacts on human health, including carcinogenesis, liver damage, kidney damage, neurological damage, immune system suppression, endocrine disruption, and fetal abnormalities (Figure 6) [272]. There has been promising use of the NPs’ unusual characteristics in biomedicine. The remarkable surface area-to-volume ratio gives NPs their unique properties. Their optical, mechanical, magnetic, and catalytic capabilities are all improved by this trait, which increases their utility in several areas, including biomedicine [273]. Chemical composition is the primary criterion for NP-type classification in the biomedical domain. Polymers, liposomes, metals, ceramics, inorganic oxides, quantum dots, and NPs based on carbon are all part of this category [274]. The significance of carbon-based NPs (CBNs) has been widely recognized. These NPs, which are mostly carbon, have several uses in biomedicine, especially in the administration of drugs and related fields [275]. Nevertheless, there are valid concerns about the biomagnification and bioaccumulation of NPs in food chains [276]. These particles go along the food chain after being ingested by a variety of organisms, potentially having an impact on humans and other higher trophic levels (Figure 6).

Tests performed on exposed personnel in welding activities by Gomes et al. (2012) [277] showed that their lungs were deposited with high levels of ultrafine particles. Regardless of these results, there are still no established protocols for determining the possible effects of NPs. A lack of comprehensive data on the origins, pathways, and possible effects of NPs on human and environmental health is a pressing concern. To better understand the effects of NPs on human and environmental health as well as the hazards they pose, it would be helpful to establish such standards and provide systematic information. NPs, used in personal care products and cosmetics, can enter the skin, making dermal toxicity examination critical. A shift in the components of the environment can influence human health; TiO_2_ NPs can cause cancer, particularly in individuals in related jobs. When exposure rates are higher than 20 mg m^−3^, health problems may arise. While other forms of NPs, such as CNTs, CeO_2_ NPs, and Ag NPs, have been connected to lung malignancies, a high concentration of inhaled NPs in the respiratory tract has been associated with fibrosis.

### 7.2. Regulations for Risk Assessment and Management Related to Nanotoxicity

#### 7.2.1. Scientific-Level Regulation

The influence of NPs on the ecosystem is a major concern, and it is imperative to ensure secure disposal or recycling. Strategies and preventative actions, such as identifying NPs in trash samples and implementing sustainable production processes and greener synthesis techniques, are required to stop NP buildup. It is efficient to use a safe-by-design strategy that incorporates safety concerns from the outset of NP development. To stop contamination of the environment, regulations pertaining to hazardous wastes must be passed and treated as dangerous nano-wastes.

Approaches for managing circular waste, involving recycling of resources, source prevention, and remediation techniques for soil and water, can lessen the negative effects of NPs in the surroundings and support environmentally friendly practices. For the safe production, consumption, and removal of NPs, laws and standards tailored to their needs must be put into place. Guidelines for NP characterization, labeling, management, and secure discharge into environments should be established by regulatory organizations. Most significantly, technology-based strategies need to be designed to address the toxic effects and ubiquity of NPs in ecosystems. Adsorption of NPs is one example of an advanced engineering approach that tries to decrease NP levels and their effect on ecosystems by increasing our knowledge of their behavior and transportation in the environment. All discuss how NPs and other micropollutants can be partially or fully removed from polluted ecosystems using remediation strategies that combine physical, chemical, and biological approaches [278,279]. Filtration and adsorption are two of the reported methods for treating NPs from sewage and water [278]. Another strategy called phytoremediation takes advantage of the natural capabilities that plants must have to purify contaminated areas [280,281,282,283]. This method is both economical and environmentally friendly. Constructed wetlands (CWs), a kind of environmental sewage management technique composed of substrates, plants, and microorganisms, have been studied for their potential for cleaning up sewage carrying CuO NPs and other pollutants [284]. The research showed that after 300 days, CWs could eliminate 98.80–99.84% of CuO NPs. Notably, though, when employing CWs as environmentally friendly ways for cleaning up sewage containing CuO NPs, several aspects need to be considered. Microorganisms can potentially be used as a solution for NP remediation because of their important involvement in the buildup of micropollutants and hazardous metals [285,286]. Other physicochemical approaches that have been used to remove metals and NPs from soil include electrokinetic remediation and soil washing. By using these measures, the impact of NPs on ecosystems can be reduced.

#### 7.2.2. Organizational Level Regulations

When it comes to the specific problems and possible hazards that nanomaterials pose to human and environmental health, the existing assessment methodologies for their regulation are insufficient. Strict regulation of NPs is impeded by a lack of readily available exposure and hazard data. But things are looking up, and government agencies are working hard to resolve the problem in many nations. As an example, to comprehend the possible advantages and disadvantages of NPs, the United States government suggested allocating USD 2.1 billion (an increase of USD 201 million from the 2010 approved budget) to the multiagency national nanotechnology programs [287]. The United States and Europe are among the many nations that are shifting their stances on nanomaterial regulation. Under the US Toxic Substances Control Act (TSCA) and the Federal Insecticide, Fungicide, and Rodenticide Act (FIFRA), the EPA is primarily responsible for regulating nanomaterials. NPs are regulated by other federal agencies, such as the Food and Drug Administration (FDA). The FDA and the EPA work together on occasion to keep tabs on NPs found in food, medical devices, cosmetics, and other product categories. Nanomaterial regulation has undergone some changes at the EPA. Formerly, the Nanoscale Materials Stewardship Program (NMSP) encouraged nanomaterial manufacturers to voluntarily submit information about their products. However, to provide guidelines for the manufacture, application, and secure removal of NMs, the Environmental Protection Agency (EPA) is now requiring the collection of comprehensive data through a variety of approaches. This involves stringent guidelines for gathering data on both fresh and current NMs, as well as pre-manufacture notifications for new ones [287].

To assess the potential dangers to human and environmental health, the TSCA requests detailed information about novel chemical compounds through the pre-manufacture notification requirement. More than 160 new chemical substances, including carbon nanotubes and other nanoscale materials, have been reviewed by the EPA since 2015. The agency has taken various steps to regulate and restrict these nanomaterials, including limiting their use, mandating the use of appropriate PPE, limiting their environmental release, and conducting testing to generate data on their health and environmental effects. The TSCA restricts the production of some nanoscale materials to those that are subject to a consent order, or SNUR [288]. By requesting that manufacturers notify specifics of materials for one-time reporting and recordkeeping, the EPA is attempting to collect more thorough information on nanoscale materials under the information collection regulation. Prior to the production of these nanoscale materials, manufacturers are required to provide the EPA with information regarding the following: the amount of material to be manufactured, the method of manufacturing and processing, specifics regarding exposure and environmental discharge, and any relevant health and safety data that are available [289]. Several nanoscale pesticides are also being addressed by the EPA as part of FIFRA. To identify the nanomaterial components in pesticides, FIFA is revising pesticide registration criteria. These materials are designed to reduce or eliminate pests and germs. The FDA, like the EPA, oversees regulating nanomaterials found in a wide range of food, cosmetic, and medicinal items [290]. When it comes to cosmetics, food components, food contact compounds, and food color additives, the FDA has issued several industry recommendations [25]. Classification, Labelling, and Packaging of Substances and Mixtures (CLP) and “Registration, Evaluation, Authorization, and Restriction of Chemicals” (REACH) rules govern chemical management primarily in the European Union. Nanomaterials are not yet the subject of any specific legislation in Europe, and the REACH rules governing chemical substances fail to provide a clear distinction between nanomaterials and other chemicals. Nonetheless, European legislation regarding nanotechnology has been evolving at a rapid pace. The European Commission issued detailed guidelines for the inclusion of nanomaterials in various European legislation, including REACH and CLP, in 2011 [291]. Additionally, to make nanomaterial accounting easier for industry, the European Chemicals Agency (ECHA) and other EU Member States have released several guidance guidelines [291]. For nanomaterial safety, both Austria’s Nanotechnology Action Plan and France’s Grenelle II Act impose various reporting and tracking requirements. Nanomaterials in Canada are governed by several statutes that already exist, such as the Fertilizers, Feeds, Food, and Drugs Act, the Pest Control Products Act, and the Canadian Environmental Protection Act, 1999 (CEP). Despite this, the Canadian government is supporting and funding research on nanomaterial health and safety to reform the legislation around these materials [292]. Nanotechnology regulation and promotion programs also receive heavy funding from Asian governments. The Japanese Ministry of the Environment (MOE) and the Ministry of Health, Labor, and Welfare (MHLW) are now working on several survey reports regarding nanomaterial safety studies. In its pursuit of nanomaterial regulation, the National Institute of Advanced Industrial Science and Technology (AIST) has released crucial risk assessment data and studies on fullerenes, carbon nanotubes (CNTs), and titanium dioxide (TiO_2_), among others [292]. Several nations are implementing policies and measures to control nanomaterials, including Australia, Thailand, and Korea.

## 8. Conclusions, Challenges, and Prospects

The study looked at many interactions in the agroecosystem, including soil, plant, microbes, and different nanomaterials used. In the world of microbes and plants, most of the reactions are totally controlled in soil rhizosphere. Depending on the species of both plants and microbes, these relationships between microbes and plants under different soils can be positive or negative on such agroecosystem. This trend may differ under natural or engineered nanomaterials under this microbe–plant nexus. In the current review, carbon nanodots and nano-tellurium were reported as a case study with focus on the microbe–plant nexus. Transport, uptake, accumulation and nanotoxicity of nanomaterials in different plant organs and in soil microbes as well as suggested regulations were also discussed. From one side, NMs can affect soil physicochemical properties, microbial activities and plants growth, many hot topics still need more investigations such as impacts of NMs-microbes, plants nexus on soil pollution and behavior of pollutants in soil and groundwater, nano-stress on soil–plant–microbe system and understanding these interactions for the ecological impacts on human health. Microbial engineering strategies are needed for reducing the nanotoxicity on food chain in the agroecosystems. Evaluation of NMs presence in agroecosystems and their monitoring methods are crucial in a multidisciplinary approach along with the integrated analytical methodologies and techniques. Economically, the nano-agri sector promises substantial yield increases, but it also requires significant investments. As the technology permeates the agricultural supply chain, ramifications on job markets, trade dynamics, and global competitiveness become evident. Looking forward, anticipated advancements include smart nanodevices, potent nano-bio interfaces, and self-repairing materials. Nanobots, soil health rejuvenation techniques, and advanced nano-encapsulation are among the many potential R&D avenues. The road ahead requires collaborative efforts from governments, research institutions, farmers, and the private sector. Public–private partnerships, in particular, could prove indispensable, merging public-sector oversight with private-sector innovation.

## Figures and Tables

**Figure 1 nanomaterials-14-01249-f001:**
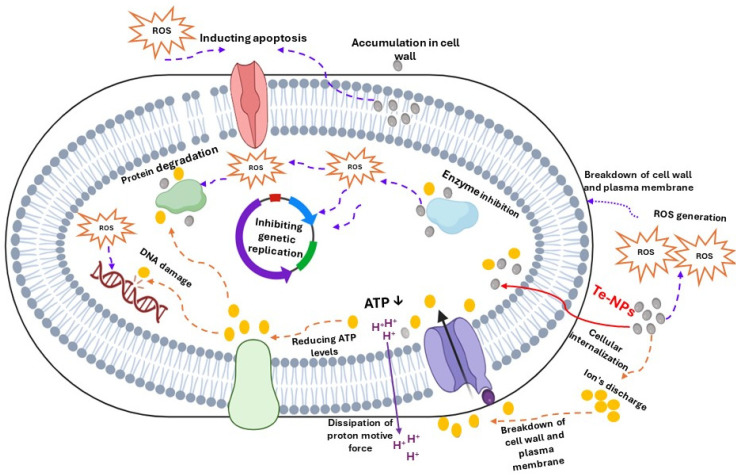
The suggested mechanism of the antimicrobial action of nano-tellurium (Te-NPs).

**Figure 2 nanomaterials-14-01249-f002:**
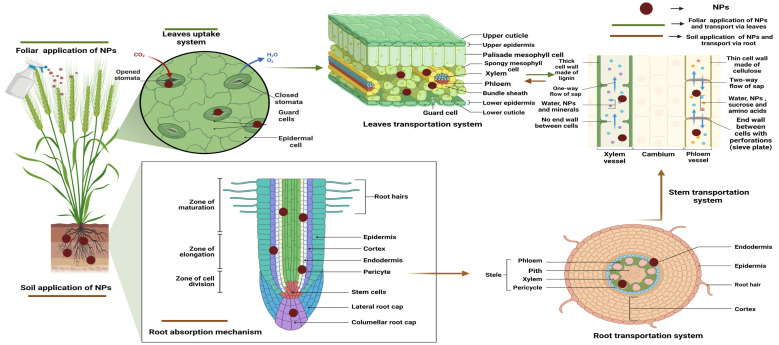
Application methodology of NPs through foliar and soil application. The NPs that apply through foliar method uptake (green color mark) by plants through stomata and soil mixture (brown color mark) NPs transport via symplastic and apoplastic pathways through xylem and phloem in different parts of plants.

**Figure 3 nanomaterials-14-01249-f003:**
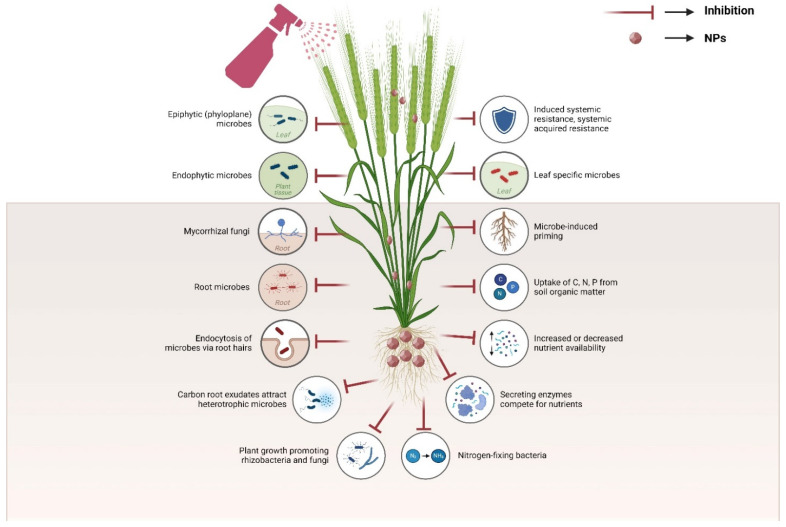
Plant and microbes’ interaction and application of NPs inhibited this positive relationship.

**Figure 4 nanomaterials-14-01249-f004:**
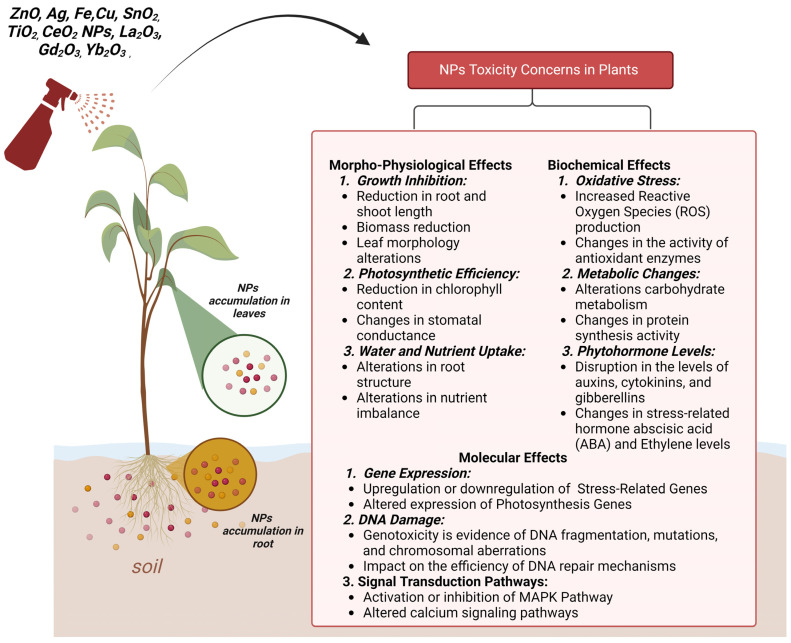
Application NPs and their negative impact on morpho-physiological, biochemical, and molecular levels. This alteration reduced the plant growth and development.

**Figure 5 nanomaterials-14-01249-f005:**
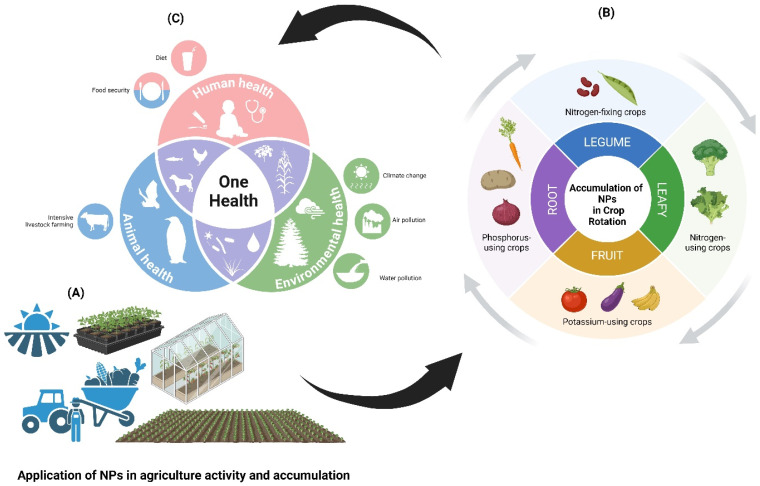
Application of NPs in agriculture, which accumulated in different crops that affected human, animal, and environmental health. (**A**) Application of NPs and its accumulation in the agriculture sector, or in the crop rotation (**B**), whereas (**C**) represents the concept of one health.

**Figure 6 nanomaterials-14-01249-f006:**
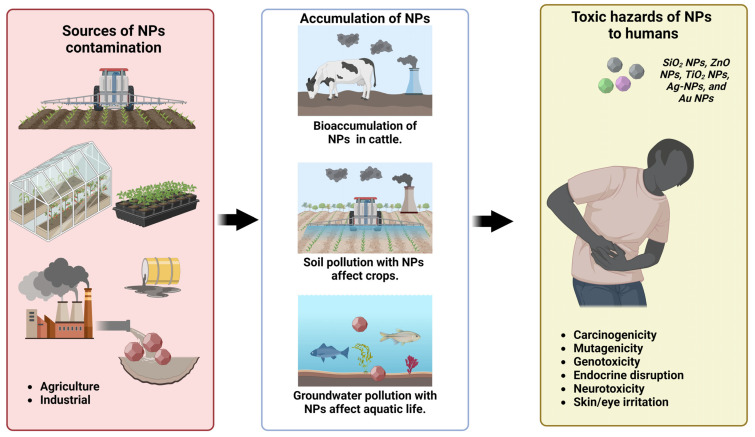
Negative impact with bioaccumulation of NPs on environment, human health.

**Table 1 nanomaterials-14-01249-t001:** Relation between microbes and plants under different soil conditions.

Plant Species	Microbe(s) or Related Item	Soil Conditions	The Type of Relationship	Refs.
Plants hyper-accumulators	*Pseudomonas aeruginosa*, *Maytenus bureaviana* and *Vibrio parahaemolyticus*	Polluted soil with heavy metals (As, Cd, Hg, Cr, Cu, Pb)	Microbe–plant assisted bioremediation	[46]
Tomato (*Solanum lycopersicum* L.)	*Firmicutes* (*Caldalkalibacillus*, *Bacillus*) and *Actinobacteria* (*Streptomyces*)	Tomato ^13^C-residue decomposition under different CO_2_ levels	Plant residue C in plant/soil/microbe system	[54]
Maize, sesame, soybean, and sweet potato	Microbial biomass (fungal and bacterial), and enzyme activities	Sandy loam soil (11.30 g kg^−1^ SOM, 5.69 pH)	Intercropping system (different 4 crops)	[47]
Rice, rice, Chinese milk vetch rotations	Bacterial-derived C proportions for decomposition of organic amendments	Organic fertilization for a 40-year field experiment	Organic amendments under long-term in paddy soils	[48]
Eggplant, maize, cucumber, pumpkin, soybean	Soil bacterial communities under weed *Ageratina adenophora* L.	Soil samples were used in measure the total phenolic content	Allelopathic system in soil and gall-forming in plant tissues	[49]
*Amaranthus hypochondriacus*	Fungus-derived biochar and salt-tolerant bacterium-plant	Cd-polluted saline-alkali soils	Myco-plant remediation of polluted soils	[51]
Crop rotations, alternative, cover, and green crops	Microbial composition, abundance, diversity, functions and services	Soil quality, soil fertility, and soil microbiome functions	Plant–soil–microbe–anthropogenic activity nexus	[50]
Grasses besides soybean, wheat and maize	Soil heterotrophic bacterial communities along with soil respiration rate	Soil OM (3%), well-drained with sandy-loam texture	Land cover types, plant residues and soil microbes	[52]
Pak choi (*Brassica chinensis* L.)	Soil microbial biomass-C, -N, -P, and microbial stoichiometry	Soil pH 5.47, SOM 1.75%, available K 65.33 mg kg^−1^	Soil microplastic pollution	[55]
Ginger (*Zingiber officinale* L.)	Soil rhizosphere microbial community, beside 16 S rRNA gene sequence	Co-polluted soil with ofloxacin and chromium	Polluted soil with antibiotics and heavy metals	[56]
*Elymus nutans*, *Kobresia humilis*, *Kobresia pygmaea*	Plant–soil nematode/bacterial and fungal linkages	Pastoral soils in alpine swampy meadows as peat bog	Grazing and plant–soil biota system	[57]
*Gymnocarpos przewalskii* L.	Arbuscular mycorrhizal fungal, fungal–bacterial ratio	Grey-brown desert soil and aeolian sandy soil	Soil microbial under hyper-arid desert	[58]
Conifer tree, broadleaf tree, and grasses	Airborne bacterial composition, urban greenspace microbiomes	Soil samples were collected using a metal soil corer	Air–phyllosphere–soil continuum	[59]

**Table 3 nanomaterials-14-01249-t003:** Synthesize CNDs from microbes.

Microbe Species	Organic Precursors	Target Applications	Refs.
*Staphylococcus aureus* and *Escherichia coli*	Potato dextrose agar, and potato dextrose broth	Using the hydrothermal method at 200 °C for 24 h to detect bacteria, dead/live microbial differentiation dyes	[76]
*Bacillus subtilis*	Tea leaves	Green synthesis of CNDs from fermentation of tea leaves with *Bacillus subtilis*	[77]
*Lactobacillus plantarum* LLC-605	*L. plantarum* LLC-605 was isolated from the traditional Chinese fermented food	Commercial dead/live microbial test dyes as a new type of anti-biofilm material by hydrothermal method at 200 °C for 24 h	[78]
*Cyanobacteria*	*Cyanobacteria* powder	CNDs was produces using hydrothermal methods having high photostability and low cytotoxicity	[75]
*Bacillus cereus* MYB41-22 cells	Yeast extract	Multicolor fluorescence bioimaging at 200 °C for 12 h	[79]
*Lactobacillus acidophilus*	Cell-free supernatant (CFS) of *L. acidophilus*	Producing functionalized nano-paper for UV and antimicrobial protective food active packaging	[80]
*E. coli* and *S. aureus*	Ampicillin sodium	Applied bioimaging with high sensitivity detecting Hg^2+^ ions in live/dead microbes at 200 °C for 6 h	[81]
*Saccharomyces cerevisiae*	1% glucose, 2% yeast extract, and pH at 5.8 for 72 h	Homogeneous N and P-doped CDs (~4.1 nm) were hydrothermal synthesized (7 h at 200 °C) as non-toxic (˂3.5 mg/mL) to produce antimicrobial bacterial nano-cellulose membrane	[82]
*Staphylococcus aureus*	Vancomycin hydrochloride (VAN)	For detect poisonous tin (Sn^4+^) using VAN-CNDs (0.899 nm) through changes in the fluorescence intensity, with antibacterial low biological toxicity	[83]
*Aspergillus flavus*	Dry fungal biomass by hydrothermal method	Applied CQDs acted as a high-harvesting agent for improving the absorption of sunlight during photosynthesis through stimulating the enzymes	[84]

**Table 4 nanomaterials-14-01249-t004:** Effects of CNDs on plant growth promotion.

Plants	CND Details	Main Findings	Refs.
Rice seeds	(10–100 ppm) for 10 days (5–10 nm)	Nano-priming rice seeds using green CNDs promoted rice growth by increasing the aroma compound due to their high content of phenolic content as antioxidants	[90]
Curcumin (*Curcuma longa* L.)	1–5 mg L^−1^ (7.34 nm) for 91 days	Plastic-derived CDs significantly enhanced enzymic antioxidants through nano-priming of seeds, besides content of chlorophyll and carbohydrate	[84]
Pea (*Pisum sativum* L.) seeds	(1.3–4.0 nm; up to 2 mg mL^−1^)	Plastic derived CDs significantly enhanced enzymic antioxidants through nano-priming of seeds, beside content of chlorophyll and carbohydrate	[91]
Tomato (*Solanum lycopersicum* L.)	1–30 mg kg^−1^ (2–8 nm) for 15 days	Functional CNDs improved tomato growth, and under drought stress by activating chlorophyll forming, osmolyte synthesis, cell division, enzymatic activation, along with soil pH, organic matter, organic carbon and soil biological activities	[92]
Tomato (*Solanum lycopersicum* L.)	8–16 mg kg^−1^ (4–15 nm) for 85 days of cultivation	Soil-applied functional CNDs ameliorated negative impacts of saline-alkali condition by up-regulation effects on soil properties (fertility, enzyme activity and decline both pH and salinity) and plant physiology (antioxidants, and nutrient uptake) along with fruit quality	[93]
*Salvia miltiorrhiza*	0.5 g in 100 mL (0.8–7.2 nm)	The presence of CNDs during plant growth enhanced the adaptability to harsh environment without a reactive oxygen species burst	[2]
Soybeans	1–50 mg kg^−1^ (for 30 days)	CNDs increased the growth of soybean plant under drought stress through the enhancement of nitrogen bioavailability	[86]
Tomato, mung beans	0.015–0.13 mg mL^−1^	CNDs improved seed germination under drought stress	[87]
Rice seedlings	50–300 μg mL^−1^ for 16 days; 2.53 nm	Mg-N co-doped CNDs significantly increased the height (22.34%) and fresh biomass (70.60%) of rice plants	[89]
*Arabidopsis thaliana*	4 mg L^−1^ (2–8 nm) for 13 days	Applied functional CNDs enhanced seedlings to be longer and stronger in their roots, bigger rosette and thicker leaves due to their easier uptake, generating more positive effects on plant	[94]
Soybean, tomato, eggplant	0.14–2.24 mg mL^−1^ (for 10 days; 5 nm)	The degradable CNDs can effectively enhance the ribulose bisphosphate carboxylase oxygenase activity and then promote the dicotyledons’ growth	[95]
Rice plants	0.14–2.24 mg mL^−1^ (for 10 days; 5 nm)	CNDs can penetrate all parts of rice plants, including the cell nuclei, which can enhance the disease resistance ability	[88]
Peanut plants	180 mg L^−1^ (2–5 nm for 25 d)	Significantly improved stress-resistant properties of plants (at 180 ppm) by increasing antioxidants of SOD, CAT, and POD, and reducing MDA content	[85]
Pumpkin	100–400 ppm (2–6 nm) for 7 days	CNDs could potentially trigger the antioxidant defense systems (SOD, CAT, and POD) in pumpkin seedlings with more impacts on roots than shoots of pumpkin plants	[96]

Abbreviations: superoxide dismutase (SOD), catalase (CAT), peroxidase (POD), and malonaldehyde (MDA).

**Table 5 nanomaterials-14-01249-t005:** Phytotoxicity concerns of CNDs on plants.

Plants	Applied Level (s)	Phytotoxicity Concerns	Refs.
*Solanum nigrum* L.	5–15 mg kg^−1^ (2–6 nm) for 60 days cultivation	Functional CNDs improved soil nano-remediation by suppressing metal translocation (Cd, Pb) to shoots, activated enzymes (SOD, POD, and glutathione peroxidase), and microbial diversity in the rhizosphere	[101]
*Arabidopsis thaliana*	24.93 and 53.55 µg mL^−1^ for 30 days	CNDs decreased the photosynthesis rates and gas exchange in plants	[97]
*Allium cepa* tubers	20 µg mL^−1^, (5–10 nm), for 24 h	Sugar-terminated CNDs were found to be non-toxic as nanofertilizers can promote the growth of *Vigna radiata* seedlings under salt stress (up to 100 mM NaCl)	[102]
Water hyacinth (*Eichhornia crassipes*)	4, 8, 16, 30 mg L^−1^ for 8 days	Functional CNDs improved removing of heavy metals (Cd, Pb) by nano-phytoremediation, regulate enzymatic levels, moderate their biotoxicity and inhibit their transfer	[103]
*Citrus maxima*	600 and 900 ppm for 10 days	CNDs can be used as repair agents to mitigate the toxicity of Cd^2+^ to plants at the level of 900 ppm by mitigating the oxidative stress and reduce transported Cd^2+^ to leaves	[99]
Wheat seedlings	50 and 75 mg L^−1^ for 10 days	The toxicity of Cd^2+^ was reduced with the uptake of CNDs by reducing Cd^2+^ uptake and increase root activity	[100]
*Arabidopsis thaliana* L.	125–1000 ppm for 7 days	The phytotoxic CNDs was 1000 ppm, which led to increased activities of glutathione reductase in roots and shoots in contrast to control and reduced the metabolites	[94]

**Table 6 nanomaterials-14-01249-t006:** Antimicrobial studies on tellurium-based nanoparticles.

Materials Related to Te-NPs	Size and Morphology	Inhibited Microbes or Anti-Microbes	Refs.
Te-nanostructures of Au-, Ag-, and Au/Ag	Nanotube size (25–30 nm) wall thickness (5–6 nm)	*E. coli*, *S. aureus*, and *S. enteritidis*	[114]
Tellurium oxide NPs	Spheres diameter ~65 nm	*S. aureus*, *K. pneumoniae*, and *E. coli*	[115]
Tellurium nanorods	Rod-shaped size (22 nm), length 185 nm	Methicillin-resistant *S. aureus*, *S. typhi* (PTCC 1609), and *P. aeruginosa* (PTTC 1574)	[116]
Te-loaded polymeric fiber	Clusters ~20 µm	*P. aeruginosa*, *S. aureus* and *E. coli*	[117]
TeO_2_-NPs sols	Spheres diameter ~55 nm	*Bacillus subtilis*, *S. enteritidis*, and *E. coli*	[118]
Lime-mediated-Te-NPs, Orange-mediated-Te-NPs, Lemon-mediated-Te-NPs	Nanorods of orange Te-NPs (50–200 nm length), cubic shape in others (100–200 nm in length)	Multidrug-resistant (MDR) *E. coli* (ATCC BAA-2471) and methicillin-resistant *S. aureus* (MRSA) (ATCC 4330)	[119]
Chitosan-fabricated Te-NPs	Spheres diameter (37 nm)	*L. monocytogenes*, *B. cereus*, and *S. aureus*	[120]
Myco-synthesized Te-NPs	Spheres diameter ~60.8 nm	*S. cerevisiae* PTCC 5269, *C. albicans* ATCC 10231, and *K. pneumoniae* ATCC 10031	[121]
Gallic acid-Te NPs	Spherical size 19.74 nm	*Staphylococcus aureus*, *Salmonella enterica*, and *Escherichia coli*	[120]
Nano-tellurium	Rod shape (size 21.4 nm)	*Staphylococcal bacteremia* and *Staphylococcus aureus*	[122]
Te–CeO_2_ nanocomposite	Nanofibrous sphere (nano wools 200 nm)	*Klebsiella pneumoniae* MTCC 3384 and *Bacillus subtilis* MTCC 441	[123]
Te-doped ZnO nanoparticles	Nano-sheet at hexagonal pattern (13 mm)	Anti-bacterial (*E. coli and S. aureus*), and antifungal (*C. albicans* and *E. salmonicolor*)	[124]
Biological Te-NPs by *Acinetobacter pittii*	Rod-shaped (60–130 nm)	*Escherichia coli* BW25113	[125]
Biological Te-NPs by *Aromatoleum* sp. CIB	Rod-shaped (200 nm)	*Aromatoleum* sp. CIB (pIZ2-0135) strain	[126]
Biological Te-NPs by *Mortierella* sp. AB1	Rod-shaped (100–500 nm)	*Shigella dysenteriae*, *E. coli*, *Enterobacter sakazakii*, and *Salmonella typhimurium*	[127]
Biological Te-NPs by *Haloferaxalexandrinus* GUSF-1	Rod-shaped (7–40 nm)	*Pseudomonas aeruginosa* ATCC 9027	[128]
Biological Te-NPs by *Gayadomonas* sp. TNPM15	Nanorods (15–23 nm)	*Fusarium oxysporum* AUMC 10313 and *Alternaria alternata* AUMC 3882	[129]

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
