# Peer review of "Carbon Nanodot–Microbe–Plant Nexus in Agroecosystem and Antimicrobial Applications"

_nanomaterials, 2024, doi:10.3390/nano14151249_

Round 1

Reviewer 1 Report

Comments and Suggestions for Authors

The present paper is a well-documented review-article and recent bibliography has been used, too.

This work constitutes a significant contribution to the field of nanomaterials research, specifically in the area of plant-microbe interactions in agroecosystems. This underscores the importance of investigating these interactions in order to ensure sustainable agriculture and environmental safety. The authors provided insight into the potential of carbon nanodots (CNDs) to enhance plant growth and stress tolerance, highlighting their capacity to improve stress resistance, nutrient availability, seed germination, and plant physiological responses. Furthermore, the authors mentioned the antimicrobial properties of tellurium nanoparticles and their potential applications in medicine. Tellurium nanoparticles have been found to hinder the growth of pathogenic bacteria, and are promising antibacterial agents.

The behavior of nanomaterials in agroecosystems is influenced by various factors including soil properties, environmental conditions, and interactions with soil microorganisms. Therefore, comprehensive monitoring and assessment of nanomaterials in agroecosystems are necessary to ensure environmental safety. Although the use of nanomaterials in agroecosystems holds promise for improving plant growth and disease management, their impact on soil microbial communities and the environment is not fully understood.

The study also discusses regulation and safety measures for nanotechnology-based agri-products, emphasizing the need for thorough regulations, risk assessments, and risk management strategies to address the risks associated with these products. However, the challenges of regulating nanotechnology and nanoplastics are significant. These include the complexity of characterizing and measuring nanomaterials, lack of standardized testing methods, and need for reliable toxicity data. The importance of interdisciplinary collaboration, transparency, and public engagement in the development of effective regulatory frameworks is also highlighted.

In my opinion, these results represent an important work that could be helpful to researchers.

The paper is well organized, easy readable and presented in a well-structured manner.

Therefore, I recommend that the authors address the following aspects to enhance the quality of their study:

1.         The authors have to expand the Introduction.

2.         The abstract states that:  “this work includes both carbon nanodots and nano-tellurium as case study.” I do not think it is eloquent to say just that, considering that the study refers to the impact of several types of nanomaterials on the environment. It sounds as if this study only refers to these two types of nanomaterials. These two types of nanoparticles were part of the study, but the study in its entirety was not just about the results obtained for these two types of nanoparticles.

3.         Figure 2 Some information in the figure is not visible.

4.         Lines 417-418 and lines 425-426 - Those two sentences seem redundant.

5.         Figure 4 - Information in figure 4 is questionable. There are types of nanoparticles for which stimulatory effects on plant growth have been obtained, and there are scientific studies to support this. In my opinion, it should be specified which types of nanoparticles this generalization in Figure 4 refers to.

6.         Figure 6 - It would be useful to specify the types of nanoparticles for which these results were obtained.

Thank you for the opportunity to review this work! Best regards!

Author Response

Dear Reviewer 1#

Author response: First of all, many thanks for your time and efforts to improve our MS, thanks again

The present paper is a well-documented review-article and recent bibliography has been used, too.

Author response: many thanks again for your comment.

This work constitutes a significant contribution to the field of nanomaterials research, specifically in the area of plant-microbe interactions in agroecosystems. This underscores the importance of investigating these interactions in order to ensure sustainable agriculture and environmental safety. The authors provided insight into the potential of carbon nanodots (CNDs) to enhance plant growth and stress tolerance, highlighting their capacity to improve stress resistance, nutrient availability, seed germination, and plant physiological responses. Furthermore, the authors mentioned the antimicrobial properties of tellurium nanoparticles and their potential applications in medicine. Tellurium nanoparticles have been found to hinder the growth of pathogenic bacteria, and are promising antibacterial agents.

Author response: many thanks again for your comment

The behavior of nanomaterials in agroecosystems is influenced by various factors including soil properties, environmental conditions, and interactions with soil microorganisms. Therefore, comprehensive monitoring and assessment of nanomaterials in agroecosystems are necessary to ensure environmental safety. Although the use of nanomaterials in agroecosystems holds promise for improving plant growth and disease management, their impact on soil microbial communities and the environment is not fully understood.

Author response: many thanks again for your comment

The study also discusses regulation and safety measures for nanotechnology-based agri-products, emphasizing the need for thorough regulations, risk assessments, and risk management strategies to address the risks associated with these products. However, the challenges of regulating nanotechnology and nano-plastics are significant. These include the complexity of characterizing and measuring nanomaterials, lack of standardized testing methods, and need for reliable toxicity data. The importance of interdisciplinary collaboration, transparency, and public engagement in the development of effective regulatory frameworks is also highlighted.

Author response: many thanks again for your comment. Yes, you are right in your vision, thanks.

In my opinion, these results represent an important work that could be helpful to researchers.

Author response: many thanks again for your comment, we appreciated this opinion and your encouragements, thanks again

The paper is well organized, easy readable and presented in a well-structured manner.

Author response: many thanks again for your comment, and your so positive feedback.

Therefore, I recommend that the authors address the following aspects to enhance the quality of their study:

Author response: many thanks again for your comment, and we followed your comments in the revised MS to be ready for publication, thanks

  1. The authors have to expand the Introduction.

Author response: many thanks again for your comment, we have improved the quality abstract according to your suggestion in the manuscript which you can find in track change file on page no 2-3 & line no 81-124.

  1. The abstract states that: “this work includes both carbon nanodots and nano-tellurium as case study.” I do not think it is eloquent to say just that, considering that the study refers to the impact of several types of nanomaterials on the environment. It sounds as if this study only refers to these two types of nanomaterials. These two types of nanoparticles were part of the study, but the study in its entirety was not just about the results obtained for these two types of nanoparticles.

Author response: many thanks again for your comment, thank you for your valuable comments corrected issues related to NPs. All changes can see in page number 2 and line number 61-72.

  1. Figure 2 Some information in the figure is not visible.

Author response: many thanks again for your comment,

We corrected the figure. All changes can see in page number 46 and line number 410.

  1. Lines 417-418 and lines 425-426 - Those two sentences seem redundant.

Author response: many thanks again for your comment

We removed one of them, thanks

  1. Figure 4 - Information in figure 4 is questionable. There are types of nanoparticles for which stimulatory effects on plant growth have been obtained, and there are scientific studies to support this. In my opinion, it should be specified which types of nanoparticles this generalization in Figure 4 refers to.

Author response: many thanks again for your comment, we corrected the figure 4 as per given studies example in manuscript also that NPs we add in this figure too. All changes is seen in track change file.

  1. Figure 6 - It would be useful to specify the types of nanoparticles for which these results were obtained.

Author response: many thanks again for your comment,

we corrected the figure as per given example of studies with citation of figure.

Thank you for the opportunity to review this work! Best regards!

Author response: many thanks again for your comment, really your comments will improve our revised MS, thanks again.

Reviewer 2 Report

Comments and Suggestions for Authors

The manuscript titled “Nanomaterial-Microbe-Plant Nexus in Agroecosystem and Antimicrobial Applications” by Prokisch, J.; et al. is a Review works where the authors outlined the state-of-the-art and most recent advances in the use of nanomaterials (with special focus on nanaparticles of different nature) to enhance the productivity and quality of soils which is a topic of pivotal importance for agriculture purposes. This is an interesting work for a certain audience specialized in this field and other stakeholders. Nevertheless, the authors need to cover some aspects (below described) before this manuscrip will be finally considered for its publication in Nanomaterials journal.

1) The authors should consider to add the term “nanotoxicity” in the keyword list.

2) “The agroecosystem is the main ecosystem in which plant, animals and other organisms (…) sustainability of agriculture sector” (lines 81-84). Could the authors provide quantitative data insights about the worldwide production burdens and the associated economic impact of the agriculture sector on society. This will significantly aid the potential readers to better understand the significance of the research devoted by the authors.

3) “Microbes and plants: amazing world” (lines 109-149). Here, the authors should add a schematic representation to highlight the key factor involved in the microbe-plant interactions. This could lead a better content comprehension of this section by the potential readers.

4) “4.1. Carbon nanodots as inhibitors of phytopathogens” (lines 164-258). This section is well-explained albeit it may be convenient to briefly discuss about the positive impact of carbon nanodots against plant pathogens and fungal organisms by silencing their RNA and inhibit their growth [1].

[1] https://doi.org/10.1080/13102818.2022.2146533

5) Then (and linked to the previous point), it should be also briefly explained the positive impact of magnetic nanomaterials [1] in the agriculture Industry sector by preserving the quality of the soils.

[2] https://doi.org/10.3390/nano13182585

[3] https://doi.org/10.3390/nano11113106

6) “5. Transport of nanomaterials to different organs” (lines 312-406). What are the pivotal factors (e.g. temperature, pH of the soil, oxygen gradient, …) to control this phenomenon? The authors should add some insights in this regard.

7) “6. Nanotoxicity on soil microbes and plants” (lines 407-line 700). What are the actions to minimize the detrimental nanotoxicity effects produced by certain nanoparticles? The authors should show the most suitable strategies to fight against this non-desirable effect.

8) “7. Regulations on Nanotoxicity” (lines 701-910). This section is perfectly explained. No actions are requested from the authors.

9) “Conclusions, challenges, and prospects” (lines 911-930). This section clearly depicts the most relevant outcomes found in this field and also some challenges to be addressed. It may be convenient to add a brief statement about the potential future action lines to pursue the topic of this research.

Comments on the Quality of English Language

The manuscript is generally well-written albeit it may be desirable a latest check to polish those final details susceptible to be improved.

Author Response

Dear Reviewer 2#

Author response: many thanks again for your comment

Comments and Suggestions for Authors

The manuscript titled “Nanomaterial-Microbe-Plant Nexus in Agroecosystem and Antimicrobial Applications” by Prokisch, J.; et al. is a Review works where the authors outlined the state-of-the-art and most recent advances in the use of nanomaterials (with special focus on nanoparticles of different nature) to enhance the productivity and quality of soils which is a topic of pivotal importance for agriculture purposes. This is an interesting work for a certain audience specialized in these field and other stakeholders. Nevertheless, the authors need to cover some aspects (below described) before this manuscript will be finally considered for its publication in Nanomaterials journal.

Author response: many thanks again for your comment and so kind words, your words really enforce our willing to the best and improvement of revised MS, thanks again.

1) The authors should consider to add the term “nanotoxicity” in the keyword list.

Author response: many thanks again for your comment,

Added to the list of keywords, thanks

2) “The agroecosystem is the main ecosystem in which plant, animals and other organisms (…) sustainability of agriculture sector” (lines 81-84). Could the authors provide quantitative data insights about the worldwide production burdens and the associated economic impact of the agriculture sector on society. This will significantly aid the potential readers to better understand the significance of the research devoted by the authors.

Author response: many thanks again for your comment

we corrected it. All changes make in track change file

It is well known that, agricultural development is one of the most powerful tools to end extreme poverty, boost shared prosperity, and feed a projected 10 billion people by 2050. Growth in the agriculture sector is 2 to 4 times more effective in raising incomes among the poorest compared to other sectors. Agriculture is also crucial to economic growth: accounting for 4% of global gross domestic product (GDP) and in some least developing countries, it can account for more than 25% of GDP.

3) “Microbes and plants: amazing world” (lines 109-149). Here, the authors should add a schematic representation to highlight the key factor involved in the microbe-plant interactions. This could lead a better content comprehension of this section by the potential readers.

Author response: many thanks again for your comment.

we corrected it. All changes make in track change file

The key factor involved in the microbe-plant interactions may include biotic and/or abiotic stresses caused by global climate change such as stress of temperature, salinity, heavy metal, greenhouse gases and drought. These stresses can control the plant-microbe interactions as a result of changing environment revealed that microbes could have both positive and negative results on plant growth and development. Abiotic and biotic stresses are equally problematic for crop plants, but research focused on plant-microbiome interactions promise for increasing their resilience and producing resistant crops.

Shree B, Jayakrishnan U and Bhushan S (2022) Impact of key parameters involved with plant-microbe interaction in context to global climate change. Front. Microbiol. 13:1008451. doi: 10.3389/fmicb.2022.1008451

4) “4.1. Carbon nanodots as inhibitors of phytopathogens” (lines 164-258). This section is well-explained albeit it may be convenient to briefly discuss about the positive impact of carbon nanodots against plant pathogens and fungal organisms by silencing their RNA and inhibit their growth [1]. [1] https://doi.org/10.1080/13102818.2022.2146533

Author response: many thanks again for your comment.

we corrected it. All changes make in track change file

Carbon quantum dots have many potential applications due to their cell-penetrating ability, biocompatibility and tunable properties. CQDs have the ability to inhibit plant pathogenic fungi such as Phytophthora infestans, Botrytis cinerea, Alternaria alternata and Fusarium oxysporum. CQDs also can improve gene silencing caused by exogenous dsRNA in P. infestans by causing a significant reduction of the transcript levels of the target gene in developing sporangia, without cytotoxicity of the CQDs, in the concentrations active against the plant pathogens [Kostov et al. 2022].

Kostov, K., Andonova-Lilova, B., & Smagghe, G. (2022). Inhibitory activity of carbon quantum dots against Phytophthora infestans and fungal plant pathogens and their effect on dsRNA-induced gene silencing. Biotechnology & Biotechnological Equipment, 36(1), 949–959. https://doi.org/10.1080/13102818.2022.2146533

5) Then (and linked to the previous point), it should be also briefly explained the positive impact of magnetic nanomaterials

[1] in the agriculture Industry sector by preserving the quality of the soils.

[2] https://doi.org/10.3390/nano13182585

[3] https://doi.org/10.3390/nano11113106

Author response: many thanks again for your comment.

On the other hand, there are many positive applications of functionalized magnetic nanomaterials in the agricultural sector, which should be explored. These functional magnetic nanomaterials based on iron, iron oxide, cobalt, cobalt and nickel ferrite nanoparticles can be applied in agriculture due to their unique and tunable magnetic properties, the existing versatility with regard to their (bio)functionalization, and in some cases, their inherent ability to increase crop yield [Spanos et al. 2021]. Such nanomaterials are promising in the agriculture industry sector by preserving the quality of the soils such as removing pollutants from soil and wastewater [Winkler et al. 2023; Gao et al.2024].

Spanos, A.; Athanasiou, K.; Ioannou, A.; Fotopoulos, V.; Krasia-Christoforou, T. Functionalized Magnetic Nanomaterials in Agricultural Applications. Nanomaterials 202111, 3106. https://doi.org/10.3390/nano11113106

Winkler, R.; Ciria, M.; Ahmad, M.; Plank, H.; Marcuello, C. A Review of the Current State of Magnetic Force Microscopy to Unravel the Magnetic Properties of Nanomaterials Applied in Biological Systems and Future Directions for Quantum Technologies. Nanomaterials 202313, 2585. https://doi.org/10.3390/nano13182585

Gao Y, Zhou L, Ouyang S, Sun J, Zhou Q. Environmental applications and risks of engineered nanomaterials in removing petroleum oil in soil. Sci Total Environ. 2024, 946:174165. doi: 10.1016/j.scitotenv.2024.174165.

6) “5. Transport of nanomaterials to different organs” (lines 312-406). What are the pivotal factors (e.g. temperature, pH of the soil, oxygen gradient, …) to control this phenomenon? The authors should add some insights in this regard.

Author response: many thanks again for your comment. We added this part to the revised MS:

There are many pivotal factors control the transport of nanomaterials to different plant organs such as desired application, targeted delivery, biocompatibility, the size and shape of NPs, plant species or crops [Sembada and Lenggoro 2024]. Concerning the environmental considerations, there are many factors including biodegradability and safety along with the environmental conditions. These conditions may include soil pH, temperature, and relative humidity, which play pivotal roles in determining the fate and impact of NPs [Wang et al. 2023].

Sembada, A.A.; Lenggoro, I.W. Transport of Nanoparticles into Plants and Their Detection Methods. Nanomaterials 202414, 131. https://doi.org/10.3390/nano14020131

Wang X, Xie H, Wang P, Yin H. Nanoparticles in Plants: Uptake, Transport and Physiological Activity in Leaf and Root. Materials (Basel). 2023;16(8):3097. doi: 10.3390/ma16083097. 

7) “6. Nanotoxicity on soil microbes and plants” (lines 407-line 700). What are the actions to minimize the detrimental nanotoxicity effects produced by certain nanoparticles? The authors should show the most suitable strategies to fight against this non-desirable effect.

Author response: many thanks again for your comment

We add the suggestion all these suggestions can be seen in track change file

8) “7. Regulations on Nanotoxicity” (lines 701-910). This section is perfectly explained. No actions are requested from the authors.

Author response: many thanks again for your comment, thanks again

9) “Conclusions, challenges, and prospects” (lines 911-930). This section clearly depicts the most relevant outcomes found in this field and also some challenges to be addressed. It may be convenient to add a brief statement about the potential future action lines to pursue the topic of this research.

Author response: many thanks again for your comment,

We corrected this section changes make in track change file with yellow marks.

Comments on the Quality of English Language

The manuscript is generally well-written albeit it may be desirable a latest check to polish those final details susceptible to be improved.

Author response: many thanks again for your comment, really your comments improved our MS and revised MS will be better to accepted by the reviewers and Editors as well